# Sustainable power generation for at least one month from ambient humidity using unique nanofluidic diode

Yong Zhang [1], Tingting Yang [1✉], Kedong Shang [1], Fengmei Guo [2], Yuanyuan Shang [2], Shulong Chang [2], Licong Cui [1], Xulei Lu [1], Zhongbao Jiang [1], Jian Zhou [1], Chunqiao Fu[1] & Qi-Chang He [1,3✉]

The continuous energy-harvesting in moisture environment is attractive for the development of clean energy source. Controlling the transport of ionized mobile charge in intelligent nanoporous membrane systems is a promising strategy to develop the moisture-enabled electric generator. However, existing designs still suffer from low output power density. Moreover, these devices can only produce short-term (mostly a few seconds or a few hours, rarely for a few days) voltage and current output in the ambient environment. Here, we show an ionic diode–type hybrid membrane capable of continuously generating energy in the ambient environment. The built-in electric field of the nanofluidic diode-type PN junction helps the selective ions separation and the steady-state one-way ion charge transfer. This directional ion migration is further converted to electron transportation at the surface of electrodes via oxidation-reduction reaction and charge adsorption, thus resulting in a continuous voltage and current with high energy conversion efficiency.

[1] Tribology Research Institute, School of Mechanical Engineering, Southwest Jiaotong University, Chengdu 610031, PR China. [2] Key Laboratory of Material Physics, Ministry of Education, School of Physics and Microelectronics, Zhengzhou University, Zhengzhou 450052, PR China. [3] MSME, Univ Gustave Eiffel, CNRS UMR 8208, F-77454 Marne-la-Vallée, France. ✉email: tingtingyang@swjtu.edu.cn; qi-chang.he@u-pem.fr

In recent years, water-based power generators have become a promising power generation technology due to the abundance, cleanliness and sustainability of water. Numerous water-powered generators using pure liquid water and aqueous solution as energy source have emerged[1–7], relying on streaming potential[8–13], dragging potential[14–16], waving potential[17–20] and triboelectric potential[1,20–22] etc. However, these devices require continuous or periodic water supplementation, which limits their installation location for practical use. As one form of water, moisture is abundantly present in air. The ubiquity of atmospheric moisture makes the development of moisture-based energy-harvesting technologies promising for solving the energy problem of low-power electronics and Internet of things (IoTs) devices[23–26]. In one early strategy, water vapor adsorption on porous carbon film with a nonhomogeneous vertical distribution of carboxy groups can generate a concentration gradient of the released $H^+$ ions[27–29]. But after a short duration of power output (~600 s)[29], the device voltage and current collapse since ionized mobile charge diffusion gradually reaches equilibrium, terminating the generation of electricity. Desirably, the power output should not be a transient phenomenon. However, a continuous power output relying on ambient atmospheric moisture remains challenging.

Some innovative chemical and structural designs have been proposed to satisfy requirements for continuous electric output. A typical strategy is based on nanofluidic devices, because confined nanospace and capillaries are sensitive to external stimuli[30–32] and can interact with water through many unique phenomena such as electric double layer coupling. For example, a power generator using protein nanowires film adopts the process of continuous exchange of water molecules at the solid interfaces to build a self-maintained moisture gradient[33]. Under the moisture gradient, nanowires with a high density of nanometre-scale pores and surface functional groups facilitate ionization and charge transfer for continuous electric output. Indeed, the open-circuit voltage ($V_{OC}$) and short-circuit current ($I_{SC}$) of around 0.5 V and 250 nA are generated. Significantly, the devices maintain a continuous $V_{OC}$ of 0.4–0.6 V for more than 2 months and a continuous current for at least 20 h before self-recharging. Transpiration-driven electrokinetic power generator adopting a hydrological cycle with the surrounding air is another example[34]. The incorporation of $CaCl_2$ to collect water vapor from the surrounding environment is crucial to acquiring a stable water supply to form the wet side of a carbon film. The water evaporation facilitates capillary flow from the wet to dry side, which induces a pseudo-streaming current. Meanwhile, a vertical setup causes the gradient distribution of $CaCl_2$ content by gravity. Thus, two asymmetries, i.e., protons and $Ca^{2+}$ ions, of the conductive nanoporous carbon surfaces are established, driving a continuous electrical output for at least 10 days. The devices exhibit maximum $V_{OC}$ (0.74 V), $I_{SC}$ (22.5 μA) and electric power (2.02 μW) when the film size is 3 cm × 9 cm × 0.12 mm.

In addition to common strategies such as chemical modification and microstructure control, some new materials have also been introduced into the field of moisture-based power generation. Polyelectrolyte, which releases free ions (such as protons) under moisture, has been explored as one type of efficient moist-electric generating material. When one side of the polyelectrolyte membrane is under constant moisture feeding, protons gradually migrate to the other side under the proton concentration gradient, offering a maximum $V_{OC}$ (0.8 V), $I_{SC}$ density (100 μA • cm$^{-2}$)[35]. However, for long-term measurement, the electric output drops back to zero after 2 days. Bilayer of polyelectrolyte film with heterogeneous distribution of charged mobile ions in moist air can extend the working time of one single device to at least 250 h with $V_{OC}$ of 0.95 V under 25% RH[36]. Large-scale integration of abundant generator units is even able to offer a $V_{OC}$ of more than 1000 V. However, the generated current output during 150 h shows an obvious decrement (40 nA at the beginning, 6 nA after 6 h, and 2 nA after 150 h).

In regard to daily electronic appliances, the life span of several months to several years is the threshold, which requires longer voltage and current output to meet applications in various fields. However, the performance summary of the existing humidity-enabled electric generator (HEEG) is shown in Supplementary Table 1, and simultaneous continuous voltage/current output for more than one month has not been realized. Therefore, obtaining membrane materials with new sustained energy conversion mechanism giving rise to high output power density and long-term stability is an urgent need. The following challenges need to be addressed: (1) protons dissociated from water or of ions should maintain stable directional migration; (2) The ion current in the humidity-sensitive materials should be continuously converted to the electronic current under the load resistance.

The design of our power source was inspired by solar cells. Solar photovoltaic power generation is a power generation method that uses the principle of photovoltaic effect to directly convert solar radiant energy into electrical energy. A typical solar cell is based on the PN junction semiconductor diode. After the photo-generated electron-hole pairs are generated in the barrier region of the PN junction, they are immediately separated by the built-in electric field. The photogenerated electrons are sent to the N zone, and the photogenerated holes are sent to the P zone, so that light energy is transformed into electrical energy. Up to now, previous HEEG devices often rely on nanofluid structure, in which positive and negative ions generated during water cycle transportation experience differential charge transfer, leading to hydrovoltaic effect. However, the selective separation of ions by only relying on Debye screening effect is limited. Inspired by the form of the semiconductor PN junction in photovoltaic device, we herein propose a HEEG device by adopting ionic diode–type PN junction in moist air, as shown in Fig. 1a. Besides, the strategy adopting nanofluidic diode–type PN junction has been verified as very feasible both by theory and experiment in hypersaline environment to build an osmotic power generator system[37,38]. But how to use the unique PN junction phenomenon in atmospheric moisture environment still remains a challenge. Herein, an ionic diode-type device consisting of a nanoporous carbon nanotubes (CNT) membrane and a nanochannel anodic aluminum oxide (AAO) membrane for moisture-based energy harvesting is developed as shown in Fig. 1b. The designed device holds the following six salient features:

(i) The built-in potential formed in the asymmetric nanofluidic junction facilitates ionization of adsorbed water molecules and one-way transfer of ion charge, thus causing power generation with less energy loss during the conversion process. Moreover, ordered AAO fluid channel lowers the transport resistance of the ion charge, thus further improving the output power.

(ii) It can generate electricity with a large amount of metal electrodes. In the device design, CNT functions as the top electrode, whereas metal behaves as the bottom electrodes. For inert metal, such as Au and carbon, there is no major redox reaction and the ion migration gives rise to electron transportation via charge adsorption. For active metal, such as Al, Zn, liquid metal, etc., electrodes participate in the partially reversible redox reactions. Similar to the electrode reactions of metal-air batteries, the overall discharge process reactions can be summarized as follows:
At the negative (anode) electrode:

$$M + xOH^- \rightarrow M(OH)_x^{y-} + (x - y)e \qquad (1)$$

where M represents the bottom electrode metal.

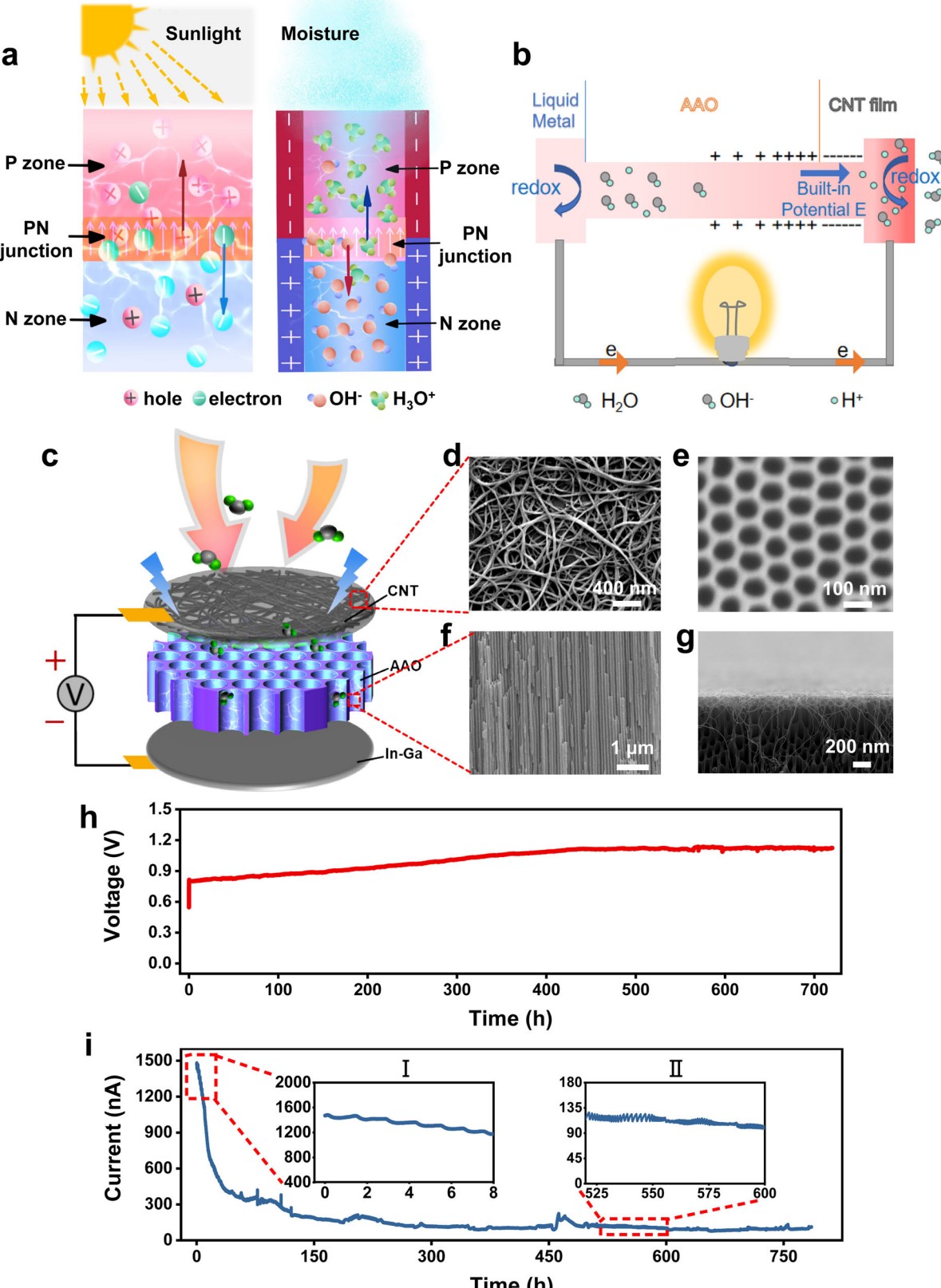

**Fig. 1 Ionic diode–type hybrid membrane device and the electric performance. a** Schematic diagram of solar photovoltaic power generation with PN junction, illustration of hydrovoltaic power generation principle inspired by photovoltaic effect. **b** Schematic diagram of the working principle for HEEG, the darker the color, the greater the humidity. **c** Explosion view of the HEEG. The SEM images of CNT top electrode (**d**), AAO from the top view (**e**) and AAO from the lateral view (**f**). **g** The SEM image of the contact interface between CNT and AAO. **h, i** The HEEG continuously work over 700 h for voltage and current respectively in 93% RH, 25 °C condition, indicating the robust and stable continuous output performance.

At the positive (cathode) electrode:

$$O_2 + 4H^+ + 4e \rightarrow 2H_2O \qquad (2)$$

This redox reaction consumes $OH^-$ and $H^+$ that are accelerated and accumulated at the two electrode ends by the built-in electric field, which converts the ionic current to electron current.

(iii) Water molecules continuously adsorb, ionize, and selectively transport due to the synergy between directed ion migration and redox reaction, guaranteeing a continuous flow of ion charge which boosts sustainable power output after an effective conversion between ion current and electron current.

(iv) The strong hydrogen bonds and van der Waals forces formed at the CNT/AAO interface endow the hybrid membrane with excellent robustness in humid environment, and such robust structure enables steady-state ion transportation and guarantees the long-term stability[39,40].

(v) The device structure is simple and the material source is wide. CNT and AAO are two of the most used materials in various fields and can be easily fabricated in large-scale. Oxygen and water in the air are cheap materials and can be supplied indefinitely without adding weight to the battery. Many common metals can be used as electrodes to participate in the discharge process, and the products do not cause any environmental pollution.

(vi) When stored in a climate-controlled facility with low humidity, ion production and transport are limited, so there is little self-discharge, ensuring a long shelf life unlike typical batteries[41].

In this work, when the bottom electrode selects liquid metal, such a nanofluidic diode-based single unit can deliver a $V_{OC}$ of 1.1 V and an $I_{SC}$ of 7.7 µA under 93% RH, 25 °C. The maximum short-circuit-current density is 11.3 µA • cm$^{-2}$ when reducing the membrane size to 1 mm$^2$, and the corresponding output power density is 1.3 µW • cm$^{-2}$/277 µW • cm$^{-3}$. Replacing the bottom electrode with a more active metal, such as Al, will further increase the $V_{OC}$ by 40% and the $I_{SC}$ by 390%. In longer-term monitoring, our fabricated membrane shows a stable $V_{OC}$ of 0.8–1.1 V for at least a month, and $I_{SC}$ undergoes gradual attenuation from 1500 nA to 100 nA after 1 month. The degradation in current is possibly associated with either the loss of oxide groups of CNT in the ambient relative humidity or the gradual passivation of bottom electrode metal. Besides, the power supply of low-power LED devices and the self-powered respiratory monitoring device have also been explored using this power generation technology.

## Results

**Design, fabrication and characterization of ionic diode–type hybrid membrane.** The experimental configuration for electricity-generating devices from ambient humidity based on ionic diode–type hybrid membrane is illustrated in Fig. 1c. The photo of the device with a working area of 0.95 cm$^2$ is shown in Supplementary Fig. 1a. As shown in the scanning electron microscope (SEM) images of the CNT film in Fig. 1d and Supplementary Fig. 1b, the CNT film prepared by chemical vapor deposition is mainly composed of interwoven carbon nanotube bundles, and the diameter of the tube bundles is mostly distributed between 10 nm and 30 nm. There are many pores in the CNT film, which allow good moisture permeability, and the pore structure of the AAO film underneath is clearly visible in the SEM photo (Supplementary Fig. 1b). AAO membrane has controllable nanochannel structures (Fig. 1e, f and Supplementary Fig. 1c, d). Compared with the messy porous structure, the oriented

nanochannel structure increases the water slip length, thereby improving the ion transport efficiency. CNT film has good electrical conductivity, gas permeability, and conformal covering ability. As the top electrode of the device, it allows for the fast passage of water molecules, and is also conducive to the efficient collection and transmission of carriers. The CNT film is relatively soft and can be closely and conformally attached to a 47-micron thick AAO substrate to form a hybrid film (Fig. 1g). Due to the contribution of capillary pressure, the high density of these nanopores helps adsorbed water molecules to form condensed liquid in the pores (Supplementary Fig. 1e, f). Supplementary Fig. 2 depicts the fabrication process of HEEG. Oxygen plasma treated carbon nanotube (CNT) thin film and liquid metal (Gallium Indium alloy) were stamped and blade-coated on the front and rear side of anodized aluminum (AAO) membrane as top and bottom electrodes, respectively. For detailed experimental procedures, please refer to the methods section. An atomic force microscope (AFM) was used to measure the thickness of the carbon nanotube film transferred to the silicon wafer (Supplementary Fig. 3). The thickness of CNT film is very thin, about tens to hundreds of nanometers. At the same time, the CNT film has excellent conductivity, and its square resistance is ~400 Ω when the thickness is 140 nm (Supplementary Fig. 4).

According to the X-ray energy dispersive spectrum (EDS) element mapping as shown in Supplementary Figs. 5 and 6, there is a large amount of oxygen distribution in AAO. Correspondingly, there should be a large number of hydroxyl groups on the surface of AAO. The high density of oxygen-containing functional groups makes it easy for moisture to penetrate into the nanochannels of AAO, and the high surface charge density increases ion selectivity. In addition, as the bottom electrode of the device, liquid metal has good conductivity and hydrophilicity. Such humidity-enabled electric generator (HEEG) can produce a continuous direct current (DC) output in a humid environment, as shown in Fig. 1h, i. Single unit with small size (0.95 cm$^2$) can deliver a continuous $V_{OC}$ of 0.8–1.1 V and a gradually decayed $I_{SC}$ (the initial maximum value is of 1.5–2.5 µA) under an environment with temperature fluctuating around 25 °C and relative humidity (RH) fluctuating around 93%. The gradually decayed current might be attributed to the inferior instability of oxygen functional groups of CNT under humid condition, as shown in Supplementary Fig. 7 and Supplementary Table 2. Another possible cause of performance degradation is the passivation of the bottom electrode metal. As shown in Supplementary Fig. 8, the device performance can be recovered to a certain extent by replacing the used bottom electrode metal with a fresh one. The role of oxygen functional groups will be discussed in detail in the mechanism section. Both $V_{OC}$ and $I_{SC}$ last for at least one month (Fig. 1h, i), indicating that the power output is not a transient phenomenon, which is quite different from porous carbon films-based hydroelectric generator with pulsed electric output.

We argue that this power generation process is related to the built-in electric field induced ionization and charge transfer effect in the CNT/AAO junction[42], as shown in Fig. 1b. Generally, carriers in semiconductor and ions in solution are similar in many ways. In nanopores, the negatively charged surface of CNT has a high concentration of positive ions which can be regarded as carriers, and is similar to the P-type semiconductor. On the contrary, the positively charged AAO channel is similar to the N-type semiconductor, and the presence of hydroxyl groups inducing the positive charged surface of AAO is verified by the Zeta potential and $I$-$V$ sweeping measurement, which will be discussed later. Joining CNT and AAO results in a PN-like junction, and in the region close to the CNT/AAO interface, the ion transport phenomenon is mainly dominated by the ionic diode-type PN junction[42]. There is a strong built-in electric field

in the depletion region of the PN junction ($\vec{E}$ from AAO to CNT), and the adsorbed water molecules partially ionize hydrogen ions and hydroxide ions under the action of the built-in electric field. The hydrogen ions move in the direction of the built-in electric field and gather near the CNT, while the hydroxide ions shift in the opposite direction. The $H^+$ and $OH^-$ accumulated at the two electrode ends by the built-in electric field can be finally consumed by a following redox reaction, which acts as the driven source of the cell. At the negative (anode) electrode, Gallium, more reactive than indium, is mainly involved in the chemical reaction. Since the chemical reaction involves multiple steps, only the typical reaction formula is given here:[43]

At the negative (anode) electrode:

$$Ga + 3OH^- \rightarrow GaO(OH) + H_2O + 3e \quad (3)$$

At the positive (cathode) electrode:

$$O_2 + 4H^+ + 4e \rightarrow 2H_2O \quad (4)$$

Therefore, if the hybrid membrane is connected to the external circuit, as long as the moisture always exists, water molecules continuously adsorb, ionize, and selectively transport due to the synergy between directed ion migration and redox reaction. For more detailed theoretical analysis, please see the supplementary material, and for more detailed experimental analysis, please refer to the mechanism section.

It is worth noting that the metal ions generated after the oxidation of the bottom electrode migrate. To evidence this migration, we replaced the liquid metal with solid Zn to avoid the disturbance caused by the liquid flow. The $Zn(OH)_4^{2-}$ formed after Zn is oxidized is easily soluble in water, which is beneficial to observing the migration phenomenon. After 25 days of continuous power generation, the distribution of Zn element is shown in Supplementary Fig. 9, indicating that chemical reaction and mass migration did occur between the Zn electrode and the electrolyte. Besides, the device was placed in a room temperature and high humid environment for cyclic voltammetry (CV) testing. After increasing the window voltage ($-1.5\,V$ to $1.5\,V$), the CV curve of the device comprises peaks indicating the redox reaction of the interface active material as shown in Supplementary Fig. 10. These experimental results provide additional evidence that redox reaction in electrode materials contributes to the electrical outputs.

**Electrical output performance of HEEG.** First, we study the influence of humidity on the HEEG performance, as shown in Fig. 2a and Supplementary Fig. 11a. With the evolution of humidity (RH of 11–40%–70–93%), more water molecules can be adsorbed, ionized, and selectively transported. Thus, $V_{OC}$ increases monotonically from 0.5 V to 1.05 V, and $I_{SC}$ augments monotonically from around 28 nA to 1.8 µA ($V_{OC}$ and $I_{SC}$ are the average values calculated during the initial working period of 2 h, the same as below). The fact that the voltage and current output augment with increasing humidity is explained by the theoretical analysis in the supplementary materials. The measured $V_{OC}$ actually consists of two parts: the built-in potential ($V_B$) generated by the power source and the redox potential ($V_{redox}$) produced by the unequal potential drop at the electrode-electrolyte interface. Once the humidity increases, the old balance is broken. More water molecules are ionized out of $H^+$ cation and $OH^-$ anion by the built-in electric field of the PN junction. Under the action of the built-in electric field, $H^+$ cation is enriched towards the CNT end, and more $OH^-$ enters the nanochannel of AAO, resulting in less $H^+$ in AAO. According to equation Eq. 1.11 in the supplementary materials, the built-in potential $V_B$ in ionic diode-type AAO/CNT junction increases with the augmentation of humidity until a new equilibrium is reached. On the other hand, according

to equation Eq. 3.2 in the supplementary materials, the redox potential $V_{redox}$ at the electrode-electrolyte interface increases or diminishes according as the activity of the reactants augments or decreases; more moisture increases the effective concentration of reactant water in favor of $V_{redox}$ improvement. Consequently, $V_{OC}$ gradually increases before reaching a new equilibrium. According to Eq. 3.3 in the supplementary materials, the short-circuit current varies with the electrode electrochemical reaction rate which is in turn proportional to the reactant concentration. Thus, the $I_{SC}$ increases also gradually until a new equilibrium is established.

This output trend contrasts with that of protein nanowire-based HEEG[33], which shows the highest $V_{OC}$ of 0.5 V at a RH of 40–50%. The influence of the thickness of CNT films on device performance is displayed in Fig. 2b. When a CNT film is too thin, the decrease of initial amount of $COO^-$ in the CNT domain leads to a lower built-in potential $V_B$ in ionic diode–type AAO/CNT junction, and the electrode resistance of the thin CNT film becomes relatively large, which is not conducive to power generation. Although the electrode resistance becomes smaller and the built-in electric field becomes larger, a CNT film that is too thick prevents water molecules from entering the AAO nanochannel. The combined effect of the above factors leads to a complex relationship between power generation performance and CNT thickness. In the experiments, as the average thickness of a CNT film increases from ~60 nm to ~1400 nm, both $V_{OC}$ and $I_{SC}$ measured show a tendency to be first increased and then saturated (Fig. 2b). The hole size of the AAO nanochannel is another important factor, as shown in Fig. 2c. Since the Debye length is related to the ion concentration in the solution, it is usually 1–100 nm. The exclusion-enrichment effect of ions requires that the smallest cross-sectional size in the nanopore channel be equal to or smaller than the Debye length[44], so too large pore size is not conducive to power generation. On the other hand, the too small pore size causes the water molecules captured by the nanopores to be relatively limited. The optimal pore size turns out to be around 90 nm.

Besides, the HEEG can be manufactured into devices with different working areas. Figure 2d and Supplementary Fig. 11b show that $V_{OC}$ varies between 0.8 and 1.1 V for different areas and $I_{SC}$ increases as the area of the device augments. When the area of a single device reaches 314 mm², short-circuit current $I_{SC}$ is ~7.7 µA. It is worth noting that the $I_{SC}$ density increases monotonically as the device area decreases, as shown in Supplementary Fig. 11c. When the device area is as small as 1 mm², the maximum short-circuit current density is 11.3 µA • cm⁻². One possible explanation is that a smaller device area is accompanied by fewer defects and a higher carrier collection efficiency of the electrode. Figure 2e, f display the power output performance under resistive load conditions for a device with 1 mm² working area. The best resistive load is 10 MΩ, and the power density is as high as 1.3 µW • cm⁻² and 277 µW • cm⁻³. In addition, the output $V_{OC}$ of the hybrid membrane has little decay after being placed in 93% RH environment for 30 days and ambient environment for another 60 days. (Supplementary Fig. 12a). Thanks to the strong hydrogen bonds and van der Waals forces at the interface between CNT films and AAO membranes, robust mechanical frameworks are provided to handle the deformation of water-related capillary and external distortion, as shown in Supplementary Fig. 12b and Supplementary Fig. 13. The device has high output power density and long-term stability at the same time, and its performance compared with devices of the same type reported in the literature is shown in Supplementary Table 1.

**Analysis of mechanism and influence factors.** To further understand the mechanism of HEEG power generation,

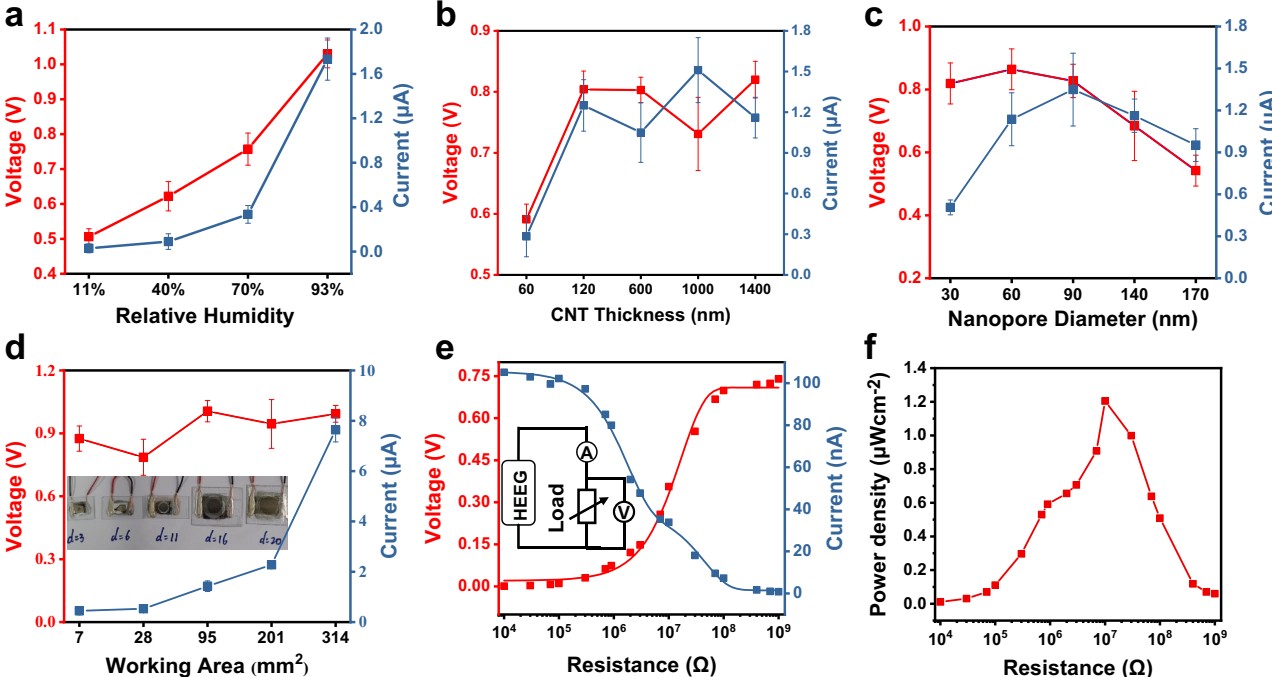

**Fig. 2 Electric output performance of HEEG and the corresponding influence factors.** Influence of relative humidity (**a**), CNT thickness (**b**), nanopore size (**c**) and device working area (**d**) to $V_{OC}$, $I_{SC}$, and the inset image in (**d**) is an optical photograph of devices with different working areas. **e** Current and voltage change with different load resistance for the device which has a working area of 1 mm². The inset is a schematic diagram of the test circuit wiring. **f** Corresponding output power density changes with different load resistance, and when external resistance is 10 MΩ, the device has the best output performance. The error bars represent standard deviation $n = 3$ independent replicates.

numerical simulations are conducted to analyze the ion transport process. Since 3D porous networks are too complex, a 2D model is used here for simplicity. CNT is negatively charged, and the negative charge should be ascribed to the carboxyl groups of CNT. AAO nanochannels contain a large number of amphoteric -OH groups, and their surface polarity is pH-dependent. The Zeta potential results show that AAO only has a positive charge density on its surface in acidic environment. In order to verify if the upper surface of the device (near the PN junction region) is in an acidic environment, we dropped the Bromothymol Blue indicator on the upper surface of the device after working in 93% RH for 12 h. The color change range of this reagent is PH 6 (yellow)−7.6 (blue). As shown in Supplementary Fig. 14, the blue indicator turned yellow after about 8 minutes, proving that the AAO near the CNT was in an acidic environment, with pH < 6. Therefore, the interfacial region of CNT/AAO acts as a nano-fluidic diode. Indeed, CNTs release free protons under moisture, leading to initial acidic environment, and the built-in electric field causes the enrichment of hydrogen ions at the CNT end, which also aggravates the acidity near the PN junction. The hydrogen ions cannot be completely consumed by the electrochemical reaction, resulting in persistent acidic conditions.

In numerical simulations, the surface charge densities of CNT and AAO are estimated as $-1.5 \, e^{-4}$ and $5 \, e^{-5}$ C/m², respectively. The pores of CNTs are taken to be of cylindrical shape with a diameter of 50 nm and a length of 1000 nm, while the channels of AAO are set as cylindrical tubes with a diameter of about 90 nm and a length of 3000 nm. Figure 3a shows the electric field distribution along the asymmetrical channel in a humid environment, the electric field direction being directed from the AAO to the CNT. Figure 3b, c show the distribution of ions concentrations along the channel under open circuit conditions. It is shown that in the PN junction region, H⁺ ions are enriched towards the CNT along the built-in electric field, and OH⁻ ions

are enriched towards the opposite direction. The H⁺ ions enrichment at the top surface of device has also been verified by the charge monitoring experiment (Supplementary Fig. 15). The results show that the plasma treatment increases the charge accumulation under the open circuit state of the device.

Under the forward voltage, the applied bias voltage is opposite to the direction of the built-in electric field formed at the AAO/CNT interface, which weakens the built-in potential. Under reverse voltage, the applied bias voltage is in the same direction as the built-in electric field formed at the AAO/CNT interface, which enhances the built-in potential. It improves the generation rate of aquatic charged ions and accelerates ion separation and directional transport. Therefore, under positive bias, the production rate of aquatic charged ions is lower and less likely to be enriched near the electrode (Supplementary Fig. 16a, b). And ions are highly enriched near the electrode under negative bias and participate in electrochemical reactions generating electronic current (Supplementary Fig. 16c, d). Consequently, the above-mentioned built-in electric field-induced ion-selective separation and transport phenomenon gives rise to a nonlinear (ionic diode-type) current in a humid environment as shown in Fig. 3d (93% RH) and 3e (11% RH).

This ionic diode-type rectification effect improves energy conversion efficiency due to steady-state one-way ion transport, and is highly dependent on surface charge density. We use oxygen plasma treatment to study the effect of surface charge density of AAO and CNT on device performance. As shown in Fig. 3f, g, plasma treatment augments the Zeta potential and the surface charge density of CNT and of AAO. Let us compare four different functional materials plasma treating methods: (i) neither CNT nor AAO with plasma treating; (ii) just AAO with air plasma treating; (iii) both with air plasma treating; (iv) both with oxygen plasma treating. Among these processing methods, both CNT and AAO with oxygen plasma treating own the best

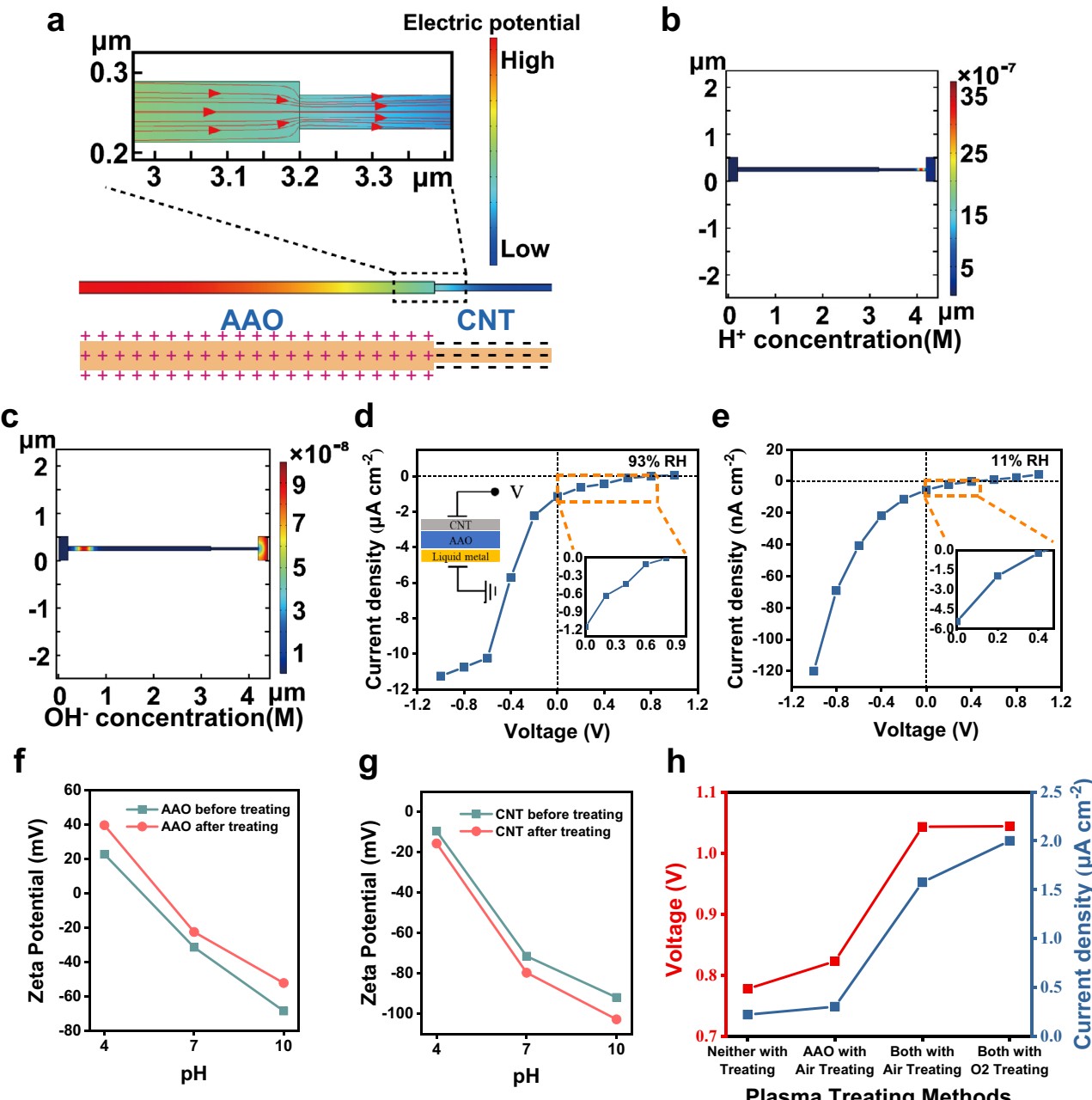

**Fig. 3 Mechanism analysis of the HEEG. a** The built-in electric field distribution under no-bias condition: the inset is a magnified view of the built-in electric field direction near the junction. **b** Concentration distribution of $H^+$ under no-bias condition: $H^+$ is transferred to the CNT side under the action of the built-in electric field. **c** $OH^-$ concentration distribution under no bias condition: $OH^-$ is transferred to the vicinity of the bottom electrode under the action of the built-in electric field. **d, e** I-V sweeping curves for device in high and low RH condition, which effectively demonstrate the ion-diode characteristics for unidirectional electrical conduction. **f, g** Surface Zeta potential of CNT film and AAO membrane in different pH condition before and after O-plasma treating, respectively, verifying effective surface modification. **h** Result of device's $V_{OC}$, $I_{SC}$ for four different plasma treating methods, and the HEEG shows the best performance after both membrane material being treated with oxygen plasma.

performance, and the $V_{OC}$ and $I_{SC}$ elevate 30% and 700% respectively compared to that of neither with plasma treating, as shown in Fig. 3h. The effect of plasma treating time has also been demonstrated as depicted in Supplementary Fig. 17. Above experimental results indicate the key role of surface charge density on the device efficiency.

The built-in electric field drives the hydroxide ions to move to the vicinity of the metal electrode so as to provide an alkaline environment and promote its redox reaction. This allows

realizing the conversion between ionic current and electronic current. Theoretically, metal with different activity undergoes different degrees of redox reactions in the electrolyte. Therefore, the output voltage and current can be adjusted not only by the changes in the surrounding humidity but also by the metal activity. To verify this conjecture, we measure the open-circuit voltage and short-circuit current of devices with a variety of bottom electrode materials, ordered from high to low activity (Standard electromotive force is shown in Supplementary

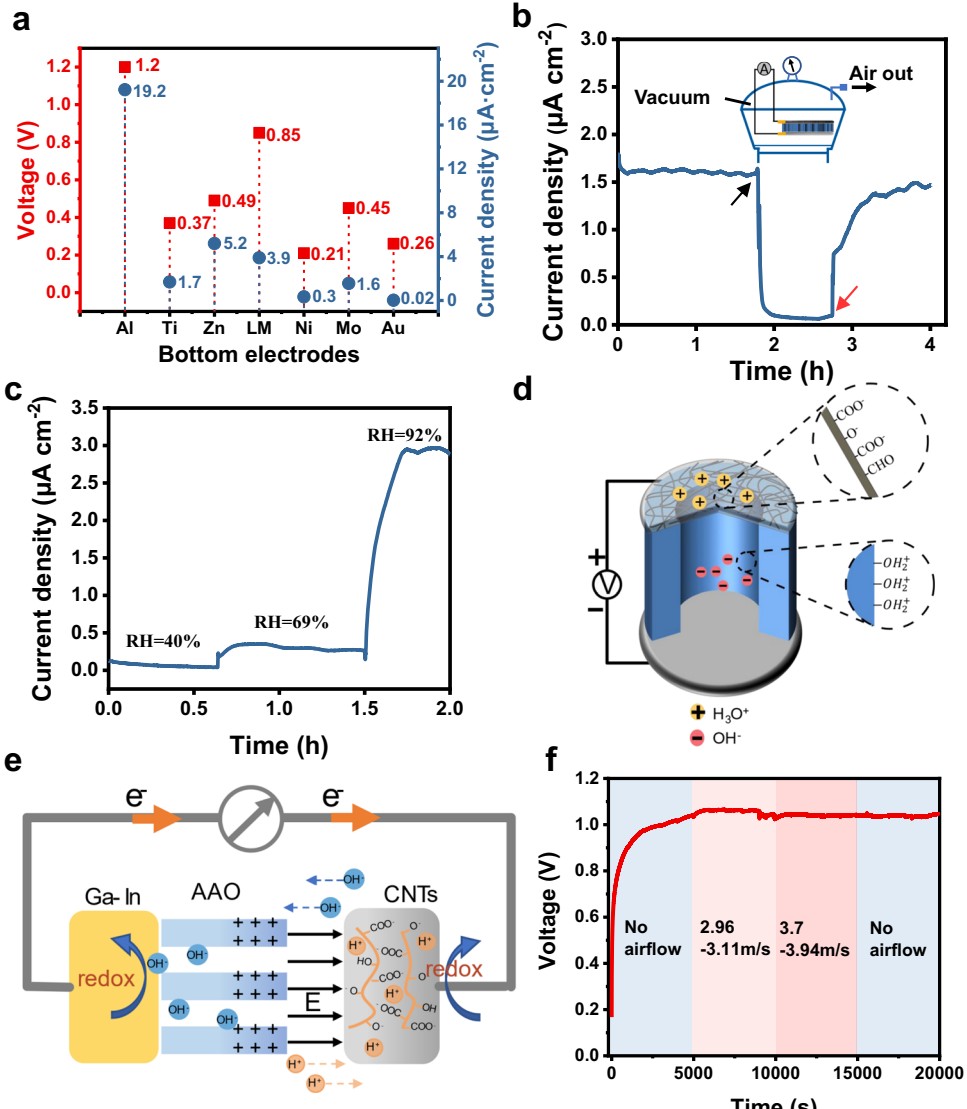

**Fig. 4 Electrical output of HEEG under different conditions. a** Performance diagram of devices made of different metal materials as bottom electrodes. **b** When the moisture (test condition 25 °C, 93% RH) in the container is pumped away (inset), the continuous current output of HEEG from the surrounding environment is destroyed (black arrow). Until the moisture is refilled, the current begins to return to its original value (red arrow). **c** The current changes continuously with the relative humidity. **d** Cartoon shows the functional groups existing on the CNT and AAO surface. **e** Schematic depiction of the harvesting moisture energy process. **f** When HEEG is exposed to the airflow environment of different wind speeds, its voltage performance has no obvious change, indicating that the airflow has little effect on the HEEG voltage signal.

Table 3), Al, Ti, Zn, Ni, Mo, Au. We keep the humidity constant at 93% RH. As shown in Fig. 4a, the $V_{OC}$ and $I_{SC}$ vary with the electrode material and are highly correlated with the electrode activity. The $V_{OC}$ and $J_{SC}$ of aluminum with the highest activity are further improved to 1.2 V and 19.2 μA • cm$^{-2}$ compared with the liquid metal of same device size. The data of some metals deviate from the activity rule, indicating that the electrical output does not completely depend on the activity. Indeed, the electrical output is also closely related to the wettability, the passivation layer, the impurity content and the real area of the electrode. Interestingly, when using the inert metal Au that does not undergo redox reactions at all, there is still a considerable voltage signal output (Au ~ 0.26 V) due to charge adsorption, which further verifies the effect of the built-in electric field-induced directional ion migration on HEEG power generation.

In addition, the use of vacuum dehumidification to prevent the dynamic adsorption of water molecules at the AAO/CNT interface depletes the electrical output, while the removal of dehumidification restores the continuous output, as shown in Fig. 4b. By increasing the relative humidity, the environment provides more water molecules, and the electrical output of the device is correspondingly increased, as shown in Fig. 4c, evidencing that the environmental humidity enables the energy generation of the device.

According to the aforementioned results, a feasible mechanism of electricity generation is illustrated in Fig. 4d, e. Since the top electrode area is exposed to moisture air and the bottom electrode area is sealed, moisture is first adsorbed on the CNT/AAO interface and become ionized. Due to the contribution of capillary pressure, the adsorbed water molecules form a condensed liquid (electrolyte) in the nanopores to facilitate

ion transport. Under the action of an electric field (directed from AAO points to CNT) built in the depletion region of the PN junction, water molecules accelerate ionization into movable hydrogen ions and hydroxide ions. Among them, the hydrogen ions move in the direction of the built-in electric field, and the hydroxide ions move in the opposite direction. Finally, $OH^-$ and $H^+$, which are accelerated at the two electrode ends by the built-in electric field, participate in the redox reaction and are continuously consumed to supply power to the outside load. Ambient environment continuously supplies water molecules, and the successive water molecule adsorption, ionization, ion selective directional transportation and redox reaction generate a sustained electric output. The device displays a particularly efficient charge transfer for continuous electric output.

Besides, we applied airflow disturbances of different magnitudes to the device in a high-humidity environment, and observed the changes of HEEG electrical signals. The device voltage and current performance show no obvious response to varied airflow flow speeds as shown in Fig. 4f and Supplementary Fig. 18. This phenomenon indicates that our power generation mechanism is quite different from those devices which are dominated by water evaporation process. It also proves that the HEEG device has reliable working stability and excellent environmental adaptability. It also should be noted that device with "CNT-Liquid Metal" structure (i.e., without AAO membrane) and around 1 $cm^2$ working area can produce a constant current ~2.8 μA, as shown in Supplementary Fig. 19. This further proves that the electrochemical reaction plays an important role in power generation of this nanofluidic diode power generator.

**Applications of HEEG**. Connecting multiple units in series and parallel to supply power to external circuit loads is of great importance to practical applications. Here we use eighteen devices, under the ambient condition (~65% RH, 25 °C), without external energy storage device and current rectification apparatus. The series-parallel combination directly lights up ten LED lights at the same time, and can also powers the electronic clock, as shown in Fig. 5a top and middle, respectively. The circuit diagram is shown in Fig. 5a bottom. This power generation device can also be used in self-powered breathing monitoring scenarios, as shown in Fig. 5b, the inset is a partial enlarged view of the generated current signal. Regular breathing of about 0.5 Hz is reliably monitored. In addition, eight power generating units connected in series charge a 47 μF capacitor to 2.7 V within 30 min in Fig. 5c. Different voltage values achieved by connecting different numbers of device units in series exhibit a good linear relationship between the voltage and the number of series units from the fitted curve of Fig. 5d. Twenty power generating units connected in series generate a $V_{OC}$ of ~18 V.

## Discussion
In this work, we have integrated the ionic diode-type hybrid membrane design into the nanofluid energy conversion system, and demonstrated a hydroelectric power generation device that continuously generate energy in environmental moisture. Through material selection and device structure design, our devices simultaneously meet the requirements of rapid capture of water molecules, high-efficiency ionization of water molecules, selective directional transport of ions, highly effective conversion between ion current and electron current. The devices elaborated in the present work provide a DC voltage and current output with maximum $V_{OC}$ and $I_{SC}$ of up to 1.1 V and 7.7 μA for a single unit, and the maximum short circuit current density and power output

density reach 11.3 μA • $cm^{-2}$ and 1.3 μW • $cm^{-2}$/277 μW • $cm^{-3}$. Most importantly, compared with most previously reported nanostructured devices whose power generation is drastically degraded after working for a few hours or days, our hydropower devices have shown continuous electrical output for at least one month. Through experimental design and theoretical analysis, the power output of our devices has a strong correlation with the built-in electric field induced ionization and charge transfer. In addition, the problem of air flow interference in practical applications has also been considered. Commercial LED displays can be easily driven by connecting multi-hydroelectric devices in series and parallel. In addition, self-powered breathing monitoring application has been explored. Our work paves a new route to generating energy in environmental moisture, promotes a better understanding of ion transport in confined nanospace and provides a universal effective way for the power supply of low-power IoTs devices.

## Methods
**Materials preparation.** the AAO membrane with different pore size and area size (fixed thickness of ~47 μm) was purchased from Shenzhen Topmembranes Technology Co, Ltd., China. CNT films were grown by chemical vapor deposition (CVD). The In-Ga liquid metal was purchased from Dongguan Wochang Metal Products Co, Ltd., China. The aluminum, titanium, zinc, nickel, molybdenum and gold targets were purchased from Deyang ONA new materials Co., Ltd., China. The Bromothymol Blue indicator was purchased online.

**Fabrication process of HEEG.** First, the commercial AAO membrane was ultrasonically cleaned with acetone for 3 min, then rinsed with deionized water for 30 s and dried with nitrogen, and then oxygen plasma treatment (MIT Corporation PCE-6) was performed for a period of time (such as 180 s, 35 W power) for standby. Second, liquid metal, as the bottom electrode, was scraped on a glass slide of appropriate size to make it evenly distributed. The AAO membrane was attached to the surface of the liquid metal (For solid metal electrode materials: use a magnetron sputtering apparatus (Automatic Magnetron Sputtering Coater, Ck300, CN), the power output is 50~100 W, spraying time is 25 min, and the thickness of the sputtered metal layer is uniformly controlled at ~120 nm, The metal lower electrode is then connected to the copper wire using silver paint, and the entire metal lower electrode area is sealed on a clean, dry glass slide using scotch tape). Third, the CNT film (~600 nm for thickness) was transferred on the upper side of the aluminum oxide membrane as the top electrode, and then silver paint was used to connect the edge of the CNT film with the copper wire. Use epoxy resin to encapsulate the silver paint part after the silver paint has dried. Finally, the prepared device was treated with high power (35 W) oxygen plasma for a period of time such as 180 s.

**Electrical output measurement and characterization.** The ion diode-type nanofluidic device output $V_{OC}$, $I_{SC}$ and the CNT film square resistance were recorded and swept by semiconductor parameter analyzer (Keithley 4200A-SCS). The humidity of the test environment was recorded by a Retronic hygrometer (Rotronic Hygrolog NT, Switzerland) with a sampling interval of 5 s. The cyclic voltammetry curve of the device in the electrochemical reaction verification test was scanned using CHI electrochemical analyzer (CHI760E, CH Instruments, Inc, US). In application section, the voltage across the capacitor was recorded in real time by an electrometer (Keithley 6514). The thickness of CNT film was characterized by an atomic force microscope (MFP-3D Infinity, Asylum Research, UK). The surface zeta potentials of the CNT film and the AAO membrane at different pH values were measured by Zeta Potential Tester (Anton Paar surpass 3, Austria) before and after the oxygen plasma treatment. Scanning electron microscope (SEM) (JSM 7800 F, Japan) was used to characterize the morphology of CNT and AAO, and X-ray energy dispersion spectroscopy (EDS) (OXFROD X-Max 80) was used to characterize the elemental composition of the hybrid membrane structure. X-ray photoelectron spectroscopy (Thermo Scientific K-Alpha) was used to analyze the surface element composition of CNT and AAO and achieve semi-quantitative analysis of their element content. Using physical vapor deposition coating technology (Automatic Magnetron Sputtering Coater, Ck300, CN) to sputter different metal materials on the bottom of AAO as bottom electrodes. A fan provides a stable and controllable wind speed and uses an anemometer to measure the wind speed (Tecman TM856). CNT film on scotch tape is photographed by an optical microscope (Axio Lab. A1, Zeiss).

**Humid environment control.** Unless otherwise stated, tests are done in a saturated salt solution environment that satisfies the humidity conditions. At a certain temperature (22 ~ 25 °C), different types of saturated salt solutions were prepared

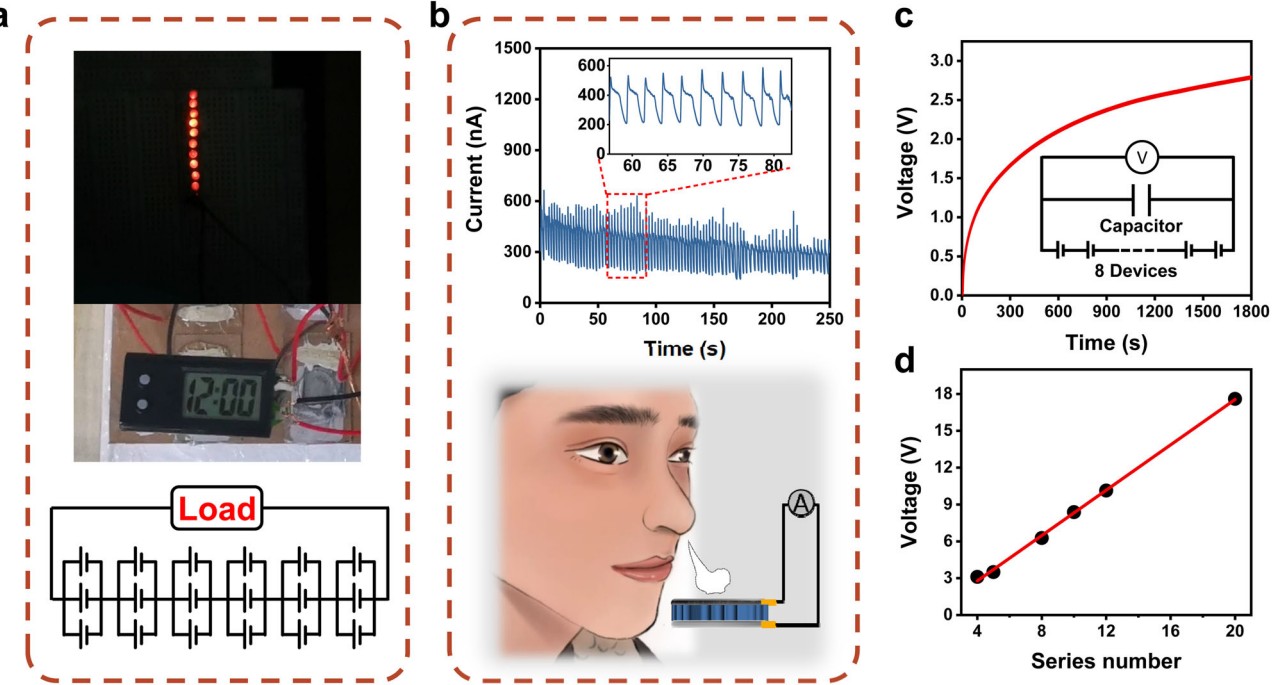

**Fig. 5 Application of HEEG devices. a** Combination of 18 devices in series and parallel connection can light up ten LEDs and power up a digital electronic watch directly under laboratory environment ~65% RH, 25 °C. **b** Self-powered breathing monitoring application and its signal display, the bottom is the test schematic diagram. **c** 8 units in series charge a 47 μF capacitor to ~2.8 V in 30 min. **d** The open-circuit voltage of different number of devices that are connected in series.

to create humid environments with different relative humidity value, as shown in supplementary table 4.

## Data availability

Source data are provided with this paper. The source dataset of this paper was uploaded to zenodo.org, https://doi.org/10.5281/zenodo.6539236, and the status is open access.

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

## Acknowledgements

We acknowledge the financial support from the National Natural Science Foundation of China (Grant Nos. 51802293, 11890673) and Fundamental Research Funds for the Central Universities of China (Grant Nos. SWJTU 2682021CX120). This work was completed at the Research Center for Ultra-precision Surface Manufacturing, Southwest Jiaotong University. We sincerely thank Linmao Qian, Wen Wang, Liang Jiang, and Bingjun Yu for providing scientifically standardized experimental sites and experimental environment, we also thank Changliu He, Pengfei Zhu, Lingxu Jia for their daily discussion. Thanks to Huiling Feng for the beautiful schematic illustrations.

## Author contributions

Y.Z. and T.Y. designed the research. T.Y. and Q.C.H. directed this research and wrote the paper. Y.Z. conducted the majority of the experimental work. T.Y. proposed mechanisms. K.S. analyzed data and discussed the results. F.G., Y.S., and S.C. prepared CNT films. L.C. tested the thickness of CNT film. X.L. and Z.J. assisted in setting up test equipment and gave suggestions on applications. J.Z., Z.B., and C.F. completed the electron micrograph shooting.

## Competing interests

The authors declare no competing interests.
