## [Peer Review File · Nature Communications]

REVIEWER COMMENTS

Reviewer #1 (Remarks to the Author):

The author fabricated a unique nanofluidic diode for electricity generation based on the ion rectification phenomenon. To verify the ion rectification phenomenon, the movement or distribution of ions within the CNT and AAO must be confirmed. Moreover, the relationship between ion current and electron current should be explained more clearly. However, the author only mentioned in the manuscript as follows, "It is still not clear how the ion current inside the device and the electronic current in the load resistance can be converted to each other". The presented results in this manuscript are also insufficient to prove the ion rectification phenomenon. In the "Analysis of mechanism and influence factors" part, the author showed electrical property by applying forward/reverse bias to the device. Based on the above results, the phenomenon of "Debye screening effect" or "pores are squeezed out against the direction of the applied bias" in the device, which was argued by the author, could not be completely demonstrated. Thus, the author should give more specific results or references to comprehensively verify the mechanism. Besides, there are many ambiguous sentences throughout the manuscript. Overall, I think the current status of this manuscript is not suitable for publication in a high-quality journal like "Nature Communications".

Additional comments are as follows:

Comment 1. Why did the author utilize In-Ga as the electrode?

Comment 2. As shown in Fig. 4c, when using an Au electrode, the generated electricity was decreased. Regarding this result, the author mentioned that the work function of Au did not match with other materials (CNT, AAO). What do you think about using other electrodes which have a similar work function with the In-Ga electrode?

Comment 3. The electrical energy did not produce when only using CNT in the device. I doubt that the result was caused by the thickness of the CNT. Please do an additional experiment by increasing the thickness of the CNT.

Comment 4. As the author mentioned that the surface zeta potential of AAO near the CNT was positive. The reason was that the H⁺ diffused from CNT, thus the AAO near the CNT was in an acidic environment. This interpretation is consistent with the result in Fig. 3c. However, why was the surface zeta potential of AAO far from the CNT also positive even though it was not affected by H⁺?

Comment 5. From the result in Fig. 4d, the Voc did not change in different airflow conditions. What about the Isc in the same conditions?

Reviewer #2 (Remarks to the Author):

This paper by Zhang and coworkers reports nanofluidic energy conversion system that continuously collects electrical energy from ambient moisture. The energy conversion system is composed of carbon nanotube (CNT)/anodic aluminum oxide (AAO) hybrid membrane that exhibits ionic current rectification (ICR) (i.e., non-linear ionic current-voltage characteristics). The optimized device provides a maximum Voc of 1.1 V under 93% RH at 25 °C, short circuit current density of 11.3 $\mu\text{A}/\text{cm}^2$ and output power density of 1.3 $\mu\text{W}/\text{cm}^2$. The devices reported in the present paper demonstrated stable and continuous electrical energy conversion for at least one month, which is a large advance from most devices reported in the literature. The authors claim that the power generation process is the consequence of the ICR phenomenon of CNT/AAO hybrid membrane, in which CNT/AAO interface functions as a diode-like pn-junction due to the p-type CNT film with negative surface charge and n-type AAO with positive surface charge. That is, the built-in electric field near the pn-junction results in ionization of water molecules and facilitates transfer of charged species.

The hydroelectric power generation from ambient moisture based on CNT/AAO hybrid membrane is of sufficient novelty, though there are several points of concern. The authors should clarify the points raised below, after which the manuscript would be suitable for Nature Communications:

1. Do the authors assume that the channels of AAO are filled with liquid water or water vapour? If the case of latter, can we discuss Debye screening length and ionic double layer (IDL)?
2. According to Figure 3c and 3d, the surface zeta potentials of CNT film and AAO membrane measured to be negative. However, the authors claim that the surface of AAO membrane is positively charged and thus the AAO channel is similar to the N-type semiconductor (line 229). The authors should clarify the origin of pn-junction due to the formation of CNT/AAO interface.
3. Figure 3a and 3b show clearly ionic current rectification at two different RHs (i.e., 93% and 11%, respectively). The devices commonly exhibit "on-state" under negative bias. That mean the forward bias condition is under the negative voltages. However, the electrode configuration (i.e., the direction of applied voltage) in the inset of Figure 3a conflicts with the measured I-V characteristics. The discussion texts in lines 321-339 need to be revised considering the accumulation and depletion of mobile charge carriers.

4. According to Figure 3a and 3b, the devices exhibit different open-circuit voltages at different RHs. The authors should discuss in detail the origin of this Voc shift.

5. The authors should support their mechanistic claims by performing numeric simulations and present the spatial distribution profiles of the electric field and the concentration of mobile ions across the hybrid membrane.

6. As Supplementary Information, the authors provided (i) analysis of built-in potential V_B in ionic diode-type AAO/CNT junction, (ii) analysis of the surface charge density of AAO and CNT, and (iii) analysis of streaming current and streaming potential in the AAO nanochannel. However, there are no qualitative discussions associated with these analysis. What is the dissociation coefficient of substance X, surface charge densities of CNT and AAO, diffusion current density, drift current, and built-in potential? and how built-in potential changes with humidity? These quantities should be provided with numbers and graphs.

7. The y-axes of Figure 3a, 3b, 3e, Figure 4a and 4b need to be "current density" rather than "current" in order to convey quantitative information.

Reviewer #3 (Remarks to the Author):

The authors proposed an ionic diode-type moisture energy harvester constructed by a hybrid carbon nanotube/anodized aluminum membrane. The structure enables long-term steady one-way ion transport and provides high energy conversion efficiency. The output power density can be as high as $1.3\mu\text{W}/\text{cm}^2$, and the cells can generate continuous power in conditions with moisture for at least one month. The first-ever strategy of adjusting the moisture-based energy harvesting performance based on the rectification effect of the ionic diode is very interesting and opens a new avenue to the nanogenerator design. Applications of the devices are also well demonstrated. However, the mechanism of the sustainable power supply of the device is not well explained. A major revision is suggested to clarify the following issues, especially the mechanism of the newly reported ionic diode power plant.

1. Can liquid metal (LM) penetrate into the AAO film and how the penetration affects the power output of the system? We can see some penetration from the EDS data (Figure S5) .
2. There is a thin layer of oxidized LM at the AAO-LM interface (due to exposure to the air or O_2 plasma), how the oxidized film on LM affects the efficiency of energy harvesting?
3. It seems that discussions about the CNT thickness effect on the performance of the generator are not correct, e.g. Line 277,". When a CNT film is too thin, the decrease of initial concentrations of COO^- in the

CNT domain leads to a lower built-in potential V_B in ionic diode-type AAO/CNT junction, and the electrode resistance of the thin CNT film becomes relatively large, which is not conducive to power generation. " Actually, the concentration of COO^- keeps constant and the resistance should decrease with the thickness of the CNT film as long the working area keeps constant. And a thick CNT film would give a low resistance intuitively.

4. The Debye length and charge screening story in the harvesting mechanism part seems not very reasonable because the EDL theory is based on the aqueous solution system, that is to say, the AAO channels are filled with water molecules, the authors are suggested to prove that the AAO channels were filled with water.

5. The mechanism of the long-term power supply of the ionic diode-based generator is not very convincing in this paper. The authors proved that no electrochemical reaction at the LM interface. However, basically, if no chemical reactions at the LM interface, the accumulation of OH^- ions at the bottom of the AAO channel would not give a sustainable electron flux to power the load, and the V_{oc} would increase as more OH^- and H^+ are accumulated at the two ends of the cell. Consumption of OH^- should be the driven source of the cell.

6. CV curves were measured to verify no electrochemical reaction at the AAO-LM interface; however, the applied voltage range is $-0.7\text{V}\sim 0.7\text{V}$ (Figure S9), I would strongly suggest extending the applied voltage range for example $\sim 1\text{V}$ (make sure no electrolysis of water, not 0.7V) to see what's new.

7. The trial of the Au electrode in line 412 proves that the voltage is due to the LM material, as a result, the use of LM may be the key to the long-term power conversion, "effective charge collection of holes and electrons" may be the reason but not very convincing.

8. Regarding monitoring electric current, I would suggest monitoring the charge in order to better understand the mechanism of the moisture-driven cell.

9. The authors claim van der Waals force and hydrogen bond between AAO and CNTs which give the robustness of the device, experiments are suggested to prove/verify.

NCOMMS-21-45536

Point-to-point response to the Reviewers' comments on the manuscript

“Sustainable power generation for at least one month from ambient humidity using unique nanofluidic diode”

We wish to thank the three Reviewers for their comments. First, we have carefully studied all these comments. Then, we have made a major revision of our initial manuscript by accounting for the comments of the three Reviewers as much as possible and in a constructive way. We believe that this revision leads to a substantial improvement of our initial manuscript.

Below, we explain how all the comments of the three Reviewers have been taken into account in our revision. Since a good few comments of the three Reviewers turn out to be quite similar, our answers to such comments are inevitably similar. **To facilitate the reading and understanding of our response to the comments of the three Reviewers, we have chosen to render each response self-contained by taking the risk of making our response appear somewhat repetitive.**

All the changes made with respect to the initial manuscript are highlighted in color in the revised manuscript.

Reviewer #1

[General comment] *The author fabricated a unique nanofluidic diode for electricity generation based on the ion rectification phenomenon. To verify the ion rectification phenomenon, the movement or distribution of ions within the CNT and AAO must be confirmed. Moreover, the relationship between ion current and electron current should be explained more clearly. However, the author only mentioned in the manuscript as follows, “It is still not clear how the ion current inside the device and the electronic current in the load resistance can be converted to each other”. The presented results in this manuscript are also insufficient to prove the ion rectification phenomenon. In the “Analysis of mechanism and influence factors” part, the author showed electrical property by applying forward/reverse bias to the device. Based on the above results, the phenomenon of “Debye screening effect” or “pores are squeezed out against the direction of the applied bias” in the device, which was argued by the author, could not be completely demonstrated. Thus, the author should give more specific results or references to comprehensively verify the mechanism. Besides, there are many ambiguous sentences throughout the manuscript. Overall, I think the current status of this manuscript is not suitable for publication in a high-quality journal like “Nature Communications”.*

[Response] After carefully studying the Reviewer's above comments, we have

carried out some additional experiments and simulations so as to account for them in revising the manuscript. Below we explain why and how the manuscript has been modified in response to the Reviewer's above comments.

- **Debye screening effect**

The concept of Debye screening length and ionic double layer (IDL) is used in the liquid case. The channels of AAO are assumed to be filled with liquid water instead of gaseous one in humid environment due to the capillary condensation effect. To check the correctness of this assumption, we scan the top surface of the device by atomic force microscopy (MFP-3D Infinity, Asylum Research, UK), and simultaneously apply moisture to the top surface of the device by bubbling method. Figures S1e and S1f show the surface topography photos of the device before and after the application of moisture. It can be seen that after the introduction of moisture (relative humidity changes from 38% RH to 60% RH), many arc bulges in the shape of water droplets are formed on the surface, proving the formation of liquid water. This confirms the correctness of our assumption.

Supplementary Figs.: 1e) The topography of the top surface area of the device swept out without moisture action(21.8°C, 38%RH); 1f) The topography of the top surface area of the device swept out after moisture action (21.8°C, 60%RH).

- **Simulation of the movement or distribution of ions within CNTs and AAOs**

To further understand the mechanism of HEEG power generation, numerical simulations are conducted to analyze the ion transport process. Since 3D porous networks are too complex, a 2D model is used here for simplicity. The surface charge densities of CNT and AAO are estimated as $-1.5e^{-4}$ and $5e^{-5}$ C/m², respectively. The pores of CNTs are taken to be of cylindrical shape with a diameter of 50 nm and a length of 1000 nm, while the channels of AAO are set as cylindrical tubes with a diameter of about 90 nm and a length of 3000 nm. Fig. 3a shows the electric field distribution along the asymmetrical channel in a humid environment, the electric field direction being directed from the AAO to the CNT. Fig. 3b and 3c display the distribution of ions concentrations along the channel under open circuit conditions. It is shown that in the PN junction region, H⁺ ions are enriched towards the CNT along the built-in electric field, and OH⁻ ions are enriched towards the opposite direction.

Fig. 3. Mechanism analysis of the HEEG. a) The built-in electric field distribution under no-bias condition: the inset is a magnified view of the built-in electric field direction near the junction. b) Concentration distribution of H^+ under no-bias condition: H^+ is transferred to the CNT side under the action of the built-in electric field. c) OH^- concentration distribution under no bias condition: OH^- is transferred to the vicinity of the bottom electrode under the action of the built-in electric field.

- **Conversion relationship between the ionic and electron currents**

For bottom electrode with inert metal, such as Au, there is no major redox reaction and the ion migration is converted to electron transportation via charge adsorption. In contrast, for active metal, such as Al, Zn, liquid metal, *etc.*, electrodes participate in the partially reversible redox reactions. Similar to the electrode reactions of metal-air batteries, the overall discharge process reactions can be summarized as follows:

At the negative (anode) electrode:

At the positive (cathode) electrode:

Above, M represents the bottom electrode metal. Such a redox reaction consumes OH^- and H^+ that are accelerated and accumulated at the two electrode ends by the built-in electric field, which converts the ionic current to electron current.

The role of redox reactions can be demonstrated by the following experiments. First, note that with the extension of power generation time, the metal ions generated after the oxidation of the bottom electrode show a migration phenomenon. Then, to illustrate the migration of metal ions, we replaced the liquid metal with solid Zn to avoid the disturbance caused by the liquid flow. The $Zn(OH)_4^{2-}$ formed after Zn is oxidized is easily soluble in water. This is beneficial to observing the migration phenomenon. After 25 days of continuous power generation, the distribution of Zn element is shown in Figure S9, indicating that chemical reaction and mass migration did occur between the Zn electrode and the electrolyte. Besides, the device was placed in a room temperature and high humid environment for cyclic voltammetry (CV) testing. After increasing the window voltage, the CV curve of the device exhibits peaks indicating the redox reaction of the interface active material as shown in Fig. S10. The above experimental results provide additional evidence that the redox reaction in electrode materials contributes to the observed electrical outputs.

Supplementary Fig. 9. After 25 days of continuous power generation, the element distribution map of the Zn-AAO cross-section shows that the zinc element has a tendency to diffuse towards the top electrode.

Supplementary Fig. 10. The volt-ampere characteristic curve: the device is placed in a 93%RH humidity environment, and the window voltage is -1.5V to 1.5V.

- **Discussing the experimental phenomenon of ion rectification**

Under the forward voltage, the applied bias voltage is opposite to the direction of the built-in electric field formed at the AAO/CNT interface, which weakens the built-in potential. Under the reverse voltage, the applied bias voltage is in the same direction as the built-in electric field formed at the AAO/CNT interface, which enhances the built-in potential. It improves the generation rate of aquatic charged ions and accelerates ion separation and directional transport. Therefore, ions are highly enriched near the electrode under negative bias and participate in electrochemical reactions generating electronic current (Fig. S15c, d); under positive bias, the production rate of aquatic charged ions is lower and less likely to be enriched near the electrode (Fig.S15a,b). Consequently, the above-mentioned built-in electric field-induced ion-selective separation and transport phenomenon gives rise to a

nonlinear (ionic diode-type) current in a humid environment as shown in Figs. 3d (93% RH) and 3e (11% RH).

Supplementary Fig.15 a). The concentration distribution of H^+ under positive bias: H^+ is concentrated near the junction b). The concentration distribution of OH^- under positive bias: OH^- is concentrated in the CNT film. c) H^+ concentration distribution under the action of reverse bias voltage: H^+ is concentrated on the side of the CNT film. d) The concentration distribution of OH^- ions under the action of reverse bias: OH^- is transported towards the bottom electrode. For the ionic rectification simulation, the initial concentration of the mobile ions (H^+ and OH^-) in the left reservoir is set to 10^{-10} mol/L and the right is set to 10^{-7} mol/L. The applied potential amplitude is 20 V.

[Comment 1] Why did the author utilize In-Ga as the electrode?

[Response] Initially we used In-Ga as the bottom electrode of the device, due to its good wettability, conductivity and simple process. To illustrate the effect of metal material selection on device performance, we supplement the experiments of electrodes with various metal materials. Theoretically, metal with different activity undergoes different degrees of redox reactions in the electrolyte. Therefore, the output voltage and current can be adjusted not only by the changes in the surrounding humidity but also by the metal activity. To verify this conjecture, we measure the open-circuit voltage and short-circuit current of devices with a variety of bottom electrode materials, ordered from high to low activity (Standard electromotive force is shown in Supplementary Table 4), Al, Ti, Zn, Ni, Mo, Au. We keep the humidity constant at 93% RH. As shown in Fig. 4a, the V_{OC} and I_{SC} vary with the electrode material and are highly correlated with the electrode activity. The V_{OC} and J_{SC} of aluminum with the highest activity are further improved to 1.2V and $19.2\mu A \cdot cm^{-2}$

compared with the liquid metal of same device size. The data of some metals deviate from the activity rule, indicating that the electrical output does not completely depend on the activity. Indeed, the electrical output is also closely related to the wettability, the passivation layer, the impurity content, and the real area of the electrode. Interestingly, when using the inert metal Au that does not undergo redox reactions at all, there is still a considerable voltage signal output (Au~0.26V) due to charge adsorption, which further verifies the effect of the built-in electric field-induced directional ion migration on HEEG power generation.

Figure 4a. Performance diagrams of devices made of different metal materials as bottom electrodes.

Supplementary Table 4| Standard electromotive force of different metals

electrode reaction	ϕ^0/V
$\text{Al}^{3+}+3\text{e}^{-}=\text{Al}$	-1.66
$\text{Ti}^{2+}+2\text{e}^{-}=\text{Ti}$	-1.63
$\text{Zn}^{2+}+2\text{e}^{-}=\text{Zn}$	-0.763
$\text{Ga}^{3+}+3\text{e}^{-}=\text{Ga}$	-0.549
$\text{Ni}^{2+}+2\text{e}^{-}=\text{Ni}$	-0.257
$\text{Mo}^{3+}+3\text{e}^{-}=\text{Mo}$	-0.22
$\text{Au}^{3+}+3\text{e}^{-}=\text{Au}$	1.5

[Comment 2] As shown in Fig. 4c, when using an Au electrode, the generated electricity was decreased. Regarding this result, the author mentioned that the work

function of Au did not match with other materials (CNT, AAO). What do you think about using other electrodes which have a similar work function with the In-Ga electrode?

[Response] We initially guessed that the work function matching would be an important factor affecting the power generation. But, with more experimental observation and theoretical analysis, we now tend to think that the conversion between ionic and electron current is directly related to the electrochemical reaction of the bottom electrode metal material. Therefore, the electrical output is closely related to the activity, the wettability, the passivation layer, the impurity content, and the real area of the electrode.

[Comment 3] *The electrical energy did not produce when only using CNT in the device. I doubt that the result was caused by the thickness of the CNT. Please do an additional experiment by increasing the thickness of the CNT.*

[Response] We thank the Reviewer for the above comment and suggestion allowing us to improve the device mechanism analysis. Indeed, we used a thicker carbon nanotube film (thickness of $\sim 900\text{nm}$) to verify the power generation effect of the device (CNT and LM in direct contact). The experimental results showed that the device did not generate voltage but a stable current signal of $\sim 2.8\mu\text{A}$ appeared. The results are shown in the following figures. They indicate that liquid metal undergoes redox reactions under high humidity conditions.

a) The current signal produced by the CNT-LM structure and b) A photo of the captured voltage signal showing 0V.

[Comment 4] *As the author mentioned that the surface zeta potential of AAO near the CNT was positive. The reason was that the H^+ diffused from CNT, thus the AAO near the CNT was in an acidic environment. This interpretation is consistent with the result in Fig. 3c. However, why was the surface zeta potential of AAO far from the CNT also positive even though it was not affected by H^+ ?*

[Response] : Thanks for your opinions. In order to verify if the upper surface of the device (near the PN junction region) is in an acidic environment, we dropped the BromothymolBlue indicator on the upper surface of the device after working in 93%RH for 12h. The color change range of this reagent is PH 6 (yellow)-7.6 (blue). As shown in Figure S14, the blue indicator turned yellow after about 8 minutes, proving that the AAO near the CNT was in an acidic environment, with $\text{pH} < 6$. Therefore, it is reasonable to consider the interfacial region of CNT/AAO as a nanofluidic diode. And the surface zeta potential of AAO far from the CNT is currently difficult to measure directly. In order to avoid misleading and ambiguity, we modify the working principle diagram (Fig. 1b) as follows, and the identification of the positive charge on the AAO surface far away from the CNT region has been deleted.

Supplementary Fig. 14. The blue Bromothymol Blue indicator was dropped on the surface of the device and gradually turned yellow after ~8 minutes, indicating that the surface of the device was weakly acidic.

Figure 1b. Schematic diagram of the working principle for HEEG, the darker the color, the greater the humidity.

[Comment 5] From the result in Fig. 4d, the V_{oc} did not change in different airflow conditions. What about the I_{sc} in the same conditions?

[Response] : Neither the V_{oc} nor the I_{sc} changed in different airflow conditions.

Supplementary Fig. 17. Current performance image of HEEG with and without airflow interference, which shows that airflow has no effects on HEEG current performance.

Reviewer #2

[General comment] This paper by Zhang and coworkers reports nanofluidic energy conversion system that continuously collects electrical energy from ambient moisture. The energy conversion system is composed of carbon nanotube (CNT)/anodic aluminum oxide (AAO) hybrid membrane that exhibits ionic current rectification (ICR) (i.e., non-linear ionic current-voltage characteristics). The optimized device provides a maximum V_{oc} of 1.1 V under 93% RH at 25 °C, short circuit current density of 11.3 $\mu\text{A}/\text{cm}^2$ and output power density of 1.3 $\mu\text{W}/\text{cm}^2$. The devices reported in the present paper demonstrated stable and continuous electrical energy conversion for at least one month, which is a large advance from most devices reported in the literature. The authors claim that the power generation process is the consequence of the ICR phenomenon of CNT/AAO hybrid membrane, in which CNT/AAO interface functions as a diode-like pn-junction due to the p-type CNT film with negative surface charge and n-type AAO with positive surface charge. That is, the built-in electric field near the pn-junction results in ionization of water molecules and facilitates transfer of charged species.

The hydroelectric power generation from ambient moisture based on CNT/AAO hybrid membrane is of sufficient novelty, though there are several points of concern. The authors should clarify the points raised below, after which the manuscript would

be suitable for Nature Communicaitons:

[Response] Thank you for your above comments on our work.

[Comment 1] *Do the authors assume that the channels of AAO are filled with liquid water or water vapour? If the case of latter, can we discuss Debye screening length and ionic double layer (IDL)?*

[Response] : The Debye screening length and ionic double layer (IDL) are discussed under the assumption that the channels of AAO are filled with liquid water in humid environment due to the capillary condensation effect. This assumption has turned to be correct after accomplishing additional experiments whose results are shown in Figures S1e and S1f, as explained in the response to the general comment of Reviewer #1.

[Comment 2] *According to Figure 3c and 3d, the surface zeta potentials of CNT film and AAO membrane measured to be negative. However, the authors claim that the surface of AAO membrane is positively charged and thus the AAO channel is similar to the N-type semiconductor (line 229). The authors should clarify the origin of pn-junction due to the formation of CNT/AAO interface.*

[Response] The reason for which the surface of AAO membrane is taken to be positively charged is given as follows: (i) CNTs release free protons under moisture, leading to initial acidic environment; (ii) the built-in electric field causes the enrichment of hydrogen ions at the CNT end, which also aggravates the acidity near the PN junction; (iii) the hydrogen ions cannot be completely consumed by the electrochemical reaction, resulting in persistent acidic conditions; (iv) under acidic conditions, the surface zeta potential of AAO membrane is measured to be positive, and the surface zeta potential of CNT membrane is measured to be negative, yielding the origin of pn-junction. In order to verify if the upper surface of the device (near the PN junction region) is in an acidic environment, we performed the following experiment. As shown in Figure S14, after device working in 93%RH for 12h, the BromothymolBlue indicator was dropped on the upper surface of the device. The color change range of this reagent is PH 6 (yellow)-7.6 (blue). The blue indicator turned yellow after about 8 minutes, proving that the AAO near the CNT was in an acidic environment, with $\text{pH} < 6$. Therefore, the interfacial region of CNT/AAO acts as a nanofluidic diode.

Supplementary Fig. 14. The blue bromothymol blue indicator was dropped on the surface of the device and gradually turned yellow after ~8 minutes, indicating that the surface of the device was weakly acidic.

[Comment 3] *Figure 3a and 3b show clearly ionic current rectification at two different RHs (i.e., 93% and 11%, respectively). The devices commonly exhibit "on-state" under negative bias. That mean the forward bias condition is under the negative voltages. However, the electrode configuration (i.e., the direction of applied voltage) in the inset of Figure 3a conflicts with the measured I-V characteristics. The discussion texts in lines 321-339 need to be revised considering the accumulation and depletion of mobile charge carriers.*

[Response] The electrode configuration (i.e., the direction of applied voltage) in the inset of Figure 3a does not conflict with the measured I-V characteristics. Indeed, under the forward voltage, the applied bias voltage is opposite to the direction of the built-in electric field formed at the AAO/CNT interface, which weakens the built-in potential. Under reverse voltage, the applied bias voltage is in the same direction as the built-in electric field formed at the AAO/CNT interface, which enhances the built-in potential. It improves the generation rate of aquatic charged ions, accelerates ion separation and directional transport. Therefore, ions are highly enriched near the electrode under negative bias to participate in electrochemical reactions to be converted into electronic current (Fig. S15c, d), while under positive bias the production rate of aquatic charged ions is lower and less likely to be enriched near the electrode (Fig.S15a,b). The above-mentioned built-in electric field-induced ion-selective separation and transport phenomenon give rise to a nonlinear (ionic diode-type) current in a humid environment as shown in Fig. 3d (93% RH) and 3e (11% RH). In conclusion, the electrode layout (direction of applied voltage) described in the inset in Figure 3d does not conflict with the I-V curve results measured in Figure 3d.

Supplementary Fig.15 a). The concentration distribution of H^+ under positive bias, H^+ is concentrated near the junction b). The concentration distribution of OH^- under positive bias, OH^- is concentrated in the CNT film. c) H^+ concentration distribution under the action of reverse bias voltage, H^+ is concentrated on the side of the CNT film. d) The concentration distribution of OH^- ions under the action of reverse bias, OH^- is transported towards the bottom electrode.

For the ionic rectification simulation, the initial concentration of the mobile ions(H^+ and OH^-) in the left reservoir is set to 10^{-10} mol/L and the right is set to 10^{-7} mol/L. The applied potential amplitude is 20 V.

[Comment 4] According to Figure 3a and 3b, the devices exhibit different open-circuit voltages at different RHs. The authors should discuss in detail the origin of this V_{oc} shift.

[Response] The voltage output with increasing humidity is explained by the theoretical analysis presented in the supplementary materials. The measured open-circuit voltage (V_{oc}) actually consists of two parts: the built-in potential (V_B) generated by the power source and the redox potential (V_{redox}) produced by the unequal potential drop at the electrode-electrolyte interface. Once the humidity increases, the old balance is broken, so that more water molecules are ionized out of H^+ cation and OH^- anion by the built-in electric field of the PN junction. Under the action of the built-in electric field, H^+ cation is enriched towards the CNT end, and more OH^- enters the nanochannel of AAO, resulting in less H^+ in AAO. According to equation Eq.1.11 in the supplementary materials, the built-in potential V_B in ionic diode-type AAO/CNT junction increases with augmenting humidity until a new

equilibrium is reached. On the other hand, according to equation Eq. 3.2 in the supplementary materials, the redox potential V_{redox} at the electrode-electrolyte interface increases or diminishes according as the activity of the reactants augments or decreases; more moisture increases the effective concentration of reactant water in favor of V_{redox} improvement. Consequently, V_{OC} gradually increases before reaching a new equilibrium.

At equilibrium (open-circuit) state, the built-in potential V_B is equal to $V_{\text{AAO}} - V_{\text{CNT}}$:

$$V_B = \frac{k_B T}{q} \ln \frac{[H^+]^{\text{CNT}}}{[H^+]^{\text{AAO}}} = \frac{k_B T}{q} \ln \frac{[OH^-]^{\text{AAO}}}{[OH^-]^{\text{CNT}}} \quad \text{Eq. 1.11}$$

$$V_{\text{redox}} = V^0 + \frac{RT}{nF} \ln \frac{\Pi a_{\text{Reactant}}^v}{\Pi a_{\text{product}}^{v'}} \quad \text{Eq.}$$

3.2

Above, V^0 is the standard electromotive force for redox reactions; v and v' represent the stoichiometric numbers of reactants and products; n denotes the number of electrons participating in the reaction; R , T and F symbolize the molar gas constant, temperature, and Faraday constant, respectively; a_{Reactant} and a_{product} are related to the activity, and the physical meaning of the activity corresponds to the effective concentration. Reactants include the bottom electrode metal, O_2 , H_2O , *etc.* The products contains metal oxides, metal hydroxides, hydroxyl metal complex ions, *etc.*

[Comment 5] *The authors should support their mechanistic claims by performing numeric simulations and present the spatial distribution profiles of the electric field and the concentration of mobile ions across the hybrid membrane.*

[Response] Following the Reviewer's suggestion, we have performed additional numerical simulations. The results of these simulations are shown about the spatial distribution profiles of the electric field (Figure 3a) and the concentration of mobile ions across the hybrid membrane (Figure 3b and 3c).

Figure 3a. The built-in electric field distribution under no-bias condition, the inset is a magnified view of the built-in electric field direction near the junction.

Figure 3. b) Concentration distribution of H^+ under no-bias condition, H^+ is transferred to the CNT side under the action of the built-in electric field. c) OH^- concentration distribution under no bias condition, OH^- is transferred to the vicinity of the bottom electrode under the action of the built-in electric field.

[Comment 6] *As Supplementary Information, the authors provided (i) analysis of built-in potential V_B in ionic diode-type AAO/CNT junction, (ii) analysis of the surface charge density of AAO and CNT, and (iii) analysis of streaming current and streaming potential in the AAO nanochannel. However, there are no qualitative discussions associated with these analyses. What is the dissociation coefficient of substance X, surface charge densities of CNT and AAO, diffusion current density, drift current, and built-in potential? and how built-in potential changes with humidity? These quantities should be provided with numbers and graphs.*

[Response] In the revised manuscript, due to considering the contribution of the electrode redox reaction, we have deleted the analysis of streaming current and streaming potential in the AAO nanochannel, which does not hold. Instead, we have added qualitative discussions associated with the built-in potential V_B , surface charge density of AAO and CNT, open circuit voltage V_{OC} and short circuit current I_{SC} as follows:

Qualitative discussions associated with V_B and surface charge density of AAO and CNT

According to Eq. 2.1 in the supplementary materials, the surface charge density of AAO and CNT increases with augmenting Zeta potential (ζ). More surface charge means larger initial functional group concentrations of [CNT] and [AAO]. According to Eq. 1.15 in the supplementary materials, the value of V_B thus also increases.

$$\sigma(\varphi_d) = \frac{2\varepsilon\varepsilon_0\kappa}{\beta e} \left[\sinh \frac{\beta e \varphi_d}{2} + \frac{2}{\kappa a} \tanh \frac{\beta e \varphi_d}{4} \right] \quad \text{Eq. 2.1}$$

$$V_B = \frac{K_B T}{q} \ln \frac{\sqrt{K_{AAO}[AAO]}\sqrt{K_{CNT}[CNT]}}{K_{IDL}[IDL]} = \frac{K_B T}{q} \left(\ln \frac{\sqrt{K_{AAO}K_{CNT}}}{K_{IDL}} + \ln \frac{\sqrt{[CNT][AAO]}}{[IDL]} \right) \quad \text{Eq. 1.15}$$

Qualitative discussions associated with V_B and V_{OC}

Our response to the question about the change of the V_B and V_{OC} with humidity has been provided in our one to Comment 4.

Qualitative discussions associated with I_{SC}

According to Eq. 3.3 in the supplementary materials, the short-circuit current varies with the electrode electrochemical reaction rate which is in turn proportional to the reactant concentration. Thus, the I_{SC} increases also gradually until a new equilibrium.

According to chemical kinetics, the relationship between the reaction rate v and the reaction activation energy ΔG is specified by

$$v = kc \exp\left(-\frac{\Delta G}{RT}\right) \quad \text{Eq. 3.3}$$

where k is the pre-exponential factor and c is the reactive particle concentration.

Concerning the dissociation coefficient of substance X , surface charge densities of CNT and AAO, diffusion current density, drift current, and built-in potential, their precise quantitative measurements are quite difficult and beyond the scope of the present work. In fact, these measurements require the accurate determination of the concentration and distribution of ions. However, the device in question is operating as an all-solid-state power source at different relative humidity degrees. Although water molecules are condensed into a liquid electrolyte, the role of interfacial water cannot be ignored in comparison with bulk water. The motion behavior of ions in the bulk water layer and the one in the interfacial water layer are very different. The overpotential during discharge process could also affect ion diffusion and current conversion. This makes it difficult to accurately measure the values of the aforementioned parameters experimentally or accurately estimate them with simple

formulas.

[Comment 7] *The y-axes of Figure 3a, 3b, 3e, Figure 4a and 4b need to be "current density" rather than "current" in order to convey quantitative information.*

[Response] Following the Reviewer's suggestion, we have changed the ordinate "current" to "current density" in Fig. 3a, 3b, 3e and Fig. 4a, 4b and redrawn these figures.

Reviewer #3

[General comment] *The authors proposed an ionic diode-type moisture energy harvester constructed by a hybrid carbon nanotube/anodized aluminum membrane. The structure enables long-term steady one-way ion transport and provides high energy conversion efficiency. The output power density can be as high as $1.3\mu\text{W}/\text{cm}^2$, and the cells can generate continuous power in conditions with moisture for at least one month. The first-ever strategy of adjusting the moisture-based energy harvesting performance based on the rectification effect of the ionic diode is very interesting and opens a new avenue to the nanogenerator design. Applications of the devices are also well demonstrated. However, the mechanism of the sustainable power supply of the device is not well explained. A major revision is suggested to clarify the following issues, especially the mechanism of the newly reported ionic diode power plant.*

[Response] We thank the Reviewer for the above comments. Following the Reviewer's suggestion, we have first conducted additional experiments and analyses to better understand and describe the mechanistic mechanism the sustainable power supply of our device. We have then carried out a major revision of our manuscript.

[Comment 1] *Can liquid metal (LM) penetrate into the AAO film and how the penetration affects the power output of the system? We can see some penetration from the EDS data (Figure S5).*

[Response] Yes, LM can penetrate into the AAO film. This is because the metal ions generated after the oxidation of the bottom electrode migrate. To illustrate the migration of metal ions, we replaced the liquid metal with Zn to avoid the disturbance caused by the liquid flow. The $\text{Zn}(\text{OH})_4^{2-}$ formed after Zn is oxidized is easily soluble in water, which is beneficial to observing the migration phenomenon. After 25 days of continuous power generation, the distribution of Zn element is shown in Figure S9,

indicating that chemical reaction and mass migration did occur between the Zn electrode and the electrolyte.

Supplementary Fig. 9. After 25 days of device working, the element distribution map of the Zn-AAO cross-section shows that the zinc element has a tendency to diffuse towards the top electrode.

To account for the Reviewer's comment, we speculate that the impact of the penetration on the power output of the device may originate from two aspects: (1) the metal ions penetrate toward the CNT electrode, which unexpectedly causes additional ion diffusion. The diffused ions will theoretically disturb the built-in electric field and the electrochemical reaction near the CNT electrode; (2) the penetration of metal ions may adversely affect the long-term working stability of the device, because this penetration is not reversible and may cause a short circuit between the top and bottom electrodes. The above conjectures need to be verified by more experiments in the future.

[Comment 2] *There is a thin layer of oxidized LM at the AAO-LM interface (due to exposure to the air or O₂ plasma), how the oxidized film on LM affects the efficiency of energy harvesting?*

[Response] We thank the Reviewer for the above comment bringing our attention to the passivation effect of the bottom electrode metal. We first put the device under normal laboratory humidity conditions (25°C, 45%RH) for 24 hours of working, then scraped off the bottom liquid metal electrode and finally replaced it with fresh liquid metal. The results are presented in Figure S8. It can be seen that replacing the fresh liquid metal significantly improves the current signal, indicating that the degradation of device performance over time is related to the passivation of the bottom electrode metal.

Supplementary Fig. 8. By replacing the original passivated LM with the device after 24 hours of working, the current performance of the device has been greatly improved.

[Comment 3] *It seems that discussions about the CNT thickness effect on the performance of the generator are not correct, e.g. Line 277". When a CNT film is too thin, the decrease of initial concentrations of COO⁻ in the CNT domain leads to a lower built-in potential V_B in ionic diode-type AAO/CNT junction, and the electrode resistance of the thin CNT film becomes relatively large, which is not conducive to power generation." Actually, the concentration of COO⁻ keeps constant and the resistance should decrease with the thickness of the CNT film as long the working area keeps constant. And a thick CNT film would give a low resistance intuitively.*

[Response] To render the mentioned discussions clear, we have modified the relevant sentence as follows: "When a CNT film is too thin, the decrease of initial amount of COO⁻ in the CNT domain leads to a lower built-in potential V_B in ionic diode-type AAO/CNT junction, and the electrode resistance of the thin CNT film becomes relatively large, which is not conducive to power generation. Although the electrode resistance becomes smaller and the built-in electric field becomes larger, a CNT film that is too thick prevents water molecules from entering the AAO nanochannel. The combined effect of the above factors leads to a complex relationship between power generation performance and CNT thickness. In the experiments, as the average thickness of a CNT film increases from ~60nm to ~1400nm, both V_{OC} and I_{SC} measured show a tendency to be first increased and then saturated (Fig. 2b)".

[Comment 4] *The Debye length and charge screening story in the harvesting mechanism part seems not very reasonable because the EDL theory is based on the aqueous solution system, that is to say, the AAO channels are filled with water molecules, the authors are suggested to prove that the AAO channels were filled with water.*

[Response] To make sure that the AAO channels are filled with water, we used the

atomic force microscopy (MFP-3D Infinity, Asylum Research, UK) to observe the topography of the device surface before and after moisture exposure. As shown in Figure S1 e and f, after the introduction of moisture (relative humidity changes from 38% RH to 60% RH), many arc bulges in the shape of water droplets are formed on the surface of CNT/AAO, proving the formation of liquid water.

Supplementary Fig. 1e) The topography of the top surface area of the device swept out without moisture action (21.8°C, 38%RH) and f) The topography of the top surface area of the device swept out after moisture action (21.8°C, 60%RH).

[Comment 5] *The mechanism of the long-term power supply of the ionic diode-based generator is not very convincing in this paper. The authors proved that no electrochemical reaction at the LM interface. However, basically, if no chemical reactions at the LM interface, the accumulation of OH⁻ ions at the bottom of the AAO channel would not give a sustainable electron flux to power the load, and the Voc would increase as more OH⁻ and H⁺ are accumulated at the two ends of the cell. Consumption of OH⁻ should be the driven source of the cell.*

[Response] After carrying out additional experiments, the electrochemical reaction has turned out to exist at the LM interface. The responses to Comments 6 and 7 provide evidence of electrochemical reaction. As the reviewer guessed, the consumption of OH⁻ is indeed the driven source of our device. To account for the Reviewer's comment, we have added the following paragraph in the Introduction of our manuscript:

In the device design, CNT functions as the top electrode, and metal functions as the bottom electrodes. For inert metal, such as Au and carbon, there is no major redox reaction and the ion migration is converted to electron transportation via charge adsorption. While for active metal, such as Al, Zn, liquid metal, *etc.*, electrodes participate in the partially reversible redox reactions. The overall discharge process reactions can be summarized as follows, similar to the electrode reactions of metal-air

batteries, where M represents the bottom electrode metal:

At the negative (anode) electrode:

At the positive (cathode) electrode:

such a redox reaction consumes OH^- and H^+ that are accelerated accumulated at the two electrode ends by the built-in electric field, which converts the ionic current to electron current.

[Comment 6] *CV curves were measured to verify no electrochemical reaction at the AAO-LM interface; however, the applied voltage range is -0.7V~0.7V (Figure S9), I would strongly suggest extending the applied voltage range for example ~1V (make sure no electrolysis of water; not 0.7V) to see what's new.*

[Response] Following the Reviewer's suggestion, we placed the device in a high humidity environment of 93%, and used an electrochemical workstation to scan the volt-ampere characteristic curve of the device. The voltage window was from -1.5V to 1.5V. The experiment indicated that the CV curve exhibited redox peaks.

Supplementary Figure 10. The volt-ampere characteristic curve, the device is placed in a 93%RH humidity environment, and the window voltage is -1.5V-1.5V.

[Comment 7] *The trial of the Au electrode in line 412 proves that the voltage is due to the LM material, as a result, the use of LM may be the key to the long-term power conversion, “effective charge collection of holes and electrons” may be the reason but not very convincing.*

[Response] To accounting for the Reviewer's comment and to clarify the effect of the bottom electrode, we have added the following discussion on the choice of different metals for the bottom electrode in the main text:

“The built-in electric field drives the hydroxide ions to move to the vicinity of the metal electrode to provide an alkaline environment and promote its redox reaction, thereby realizing the conversion between ionic current and electronic current. Theoretically, metal with different activity will undergo different degrees of redox

reactions in the electrolyte. Therefore, the output voltage and current can be adjusted not only by the changes in the surrounding humidity but also by the metal activity. To verify the above conjecture, we measured the open-circuit voltage and short-circuit current of devices with a variety of bottom electrode materials, ordered from high to low activity (Standard electromotive force is shown in Supplementary Table 4), Al, Ti, Zn, Ni, Mo, Au, respectively. We control the humidity constant at 93% RH. As shown in Fig. 4a, the V_{OC} and I_{SC} vary with the electrode material and are highly correlated with the electrode activity. The V_{OC} and J_{SC} of aluminum with the highest activity are further improved by 1.2V and $19.2\mu A \cdot cm^{-2}$ compared with the liquid metal of same device size. The data of some metals deviate from the activity rule, indicating that the electrical output does not completely depend on the activity. Indeed, the electrical output is also closely related to the wettability, the passivation layer, the impurity content, and the real area of the electrode of different metals. Interestingly, when using the inert metal Au that do not undergo redox reactions at all, there is still a considerable voltage signal output (Au~0.26V) due to charge adsorption, which further verifies the effect of the built-in electric field-induced directional ion migration on HEEG power generation.

Therefore, the key to the long-term power conversion is based on the synergy between directed ion migration (PN junction) and redox reaction (electrode-electrolyte interface), guaranteeing a continuous flow of ion charge which boosts sustainable power output after an effective conversion between ion current and electron current.”

Figure 4a. Performance diagram of devices made of different metal materials as bottom electrodes.

Supplementary Table 4| Standard electromotive force of different metals

electrode reaction	φ^0/V
$Al^{3+} + 3e^- = Al$	-1.66
$Ti^{2+} + 2e^- = Ti$	-1.63
$Zn^{2+} + 2e^- = Zn$	-0.763
$Ga^{3+} + 3e^- = Ga$	-0.549

$\text{Ni}^{2+} + 2\text{e}^- = \text{Ni}$	-0.257
$\text{Mo}^{3+} + 3\text{e}^- = \text{Mo}$	-0.22
$\text{Au}^{3+} + 3\text{e}^- = \text{Au}$	1.5

[Comment 8] *Regarding monitoring electric current, I would suggest monitoring the charge in order to better understand the mechanism of the moisture-driven cell.*

[Response] To account for the Reviewer's above comment, an additional experiment monitoring the charge has been performed. We used a Faraday cup and an electrometer to test the surface charge of the top surface of the device under open circuit condition (60%RH). The wiring method of the test process is shown in Fig. S18b, and the test results are shown in Fig. S18a. The surface charge density increases after oxygen plasma treatment.

Supplementary Fig. 18. Test of the amount of charge on the top surface of the device under open circuit condition (60%RH). a) the Faraday cup and electrometer are used to test the surface charge of the top surface of HEEG. After the device is treated with oxygen plasma, its surface charge density is improved to a certain extent; b) a schematic diagram of the wiring during the test process. During the test, the top surface of the device is close to the surface of the metal cup.

[Comment 9] *The authors claim van der Waals force and hydrogen bond between AAO and CNTs which give the robustness of the device, experiments are suggested to prove/verify.*

[Response] To account for the Reviewer's above comment, an additional experiment has been performed. We used scotch tape to stick and pull the upper surface of the plasma-treated and non-plasma-treated devices (controlling the thickness of the carbon tube to remain same). As shown in Fig. S13a, the color of the carbon tube

removed by the tape for the device without plasma cleaning is darker, indicating that the bonding between the CNT and AAO interface is weaker. In contrast, as shown in Fig. S13b, the color of the carbon tube removed by the tape for the device with plasma cleaning is lighter, indicating that the bonding between the CNT and AAO interface is stronger. In summary, plasma treatment brings more van der Waals force and hydrogen bond to the AAO/CNT interface, improving the device robustness.

Supplementary Fig. 13.a) and b) are the CNT films that were peeled off from the devices without plasma treating and with plasma treating using scotch tape, respectively. In Figure a, the color of the CNT film is darker, indicating that the bonding force of the AAO-CNT interface is weaker; the color of the CNT film in Figure b is lighter, indicating that the bonding force of the AAO-CNT interface is stronger.

REVIEWERS' COMMENTS

Reviewer #1 (Remarks to the Author):

Reviewer #1

[Response from General comment]

- Debye screening effect

The author proved that the surface properties change depending on the presence or absence of moisture from topography, but it is still unknown whether the Debye screening effect actually worked. However, since it is common to introduce the concept of Debye screening length in the research dealing with pore size, the author's approach can be considered a good attempt. Please include this part in the supporting information.

- Simulation of the movement or distribution of ions within CNTs and AAOs

The author verified the distribution and movement of ions through the simulation. It is a good attempt to understand and explain the mechanism of HEEG power generation and to clarify the behavior of ions in the HEEG system.

- Conversion relationship between the ionic and electron currents

The reversible redox reaction described by the author seems to have broadened the scope of the experiment as it is a reaction in an active metal, and I think this content is very important in this study. Additional evidence (Supplementary Fig. 9 and Supplementary Fig. 10) that the redox reaction of electrode materials contributes to electrical output is reliable data that can explain the mechanism by understanding the correlation between ion current and electron current.

- Discussing the experimental phenomenon of ion rectification

In the "Analysis of mechanism and influence factors" part, the author matched the electrical characteristics with forward/reverse bias applied to the device with the simulation data, resulting in a reliable discussion and valid data on the ion rectification phenomenon, which is the core of this study. Through this, the content of the manuscript has become more robust.

[Response from Comment 1]

It is impressive that the author provided a clear reason for using the In-Ga electrode. I believe that the output performance measured by different metal materials for the bottom electrode (Figure 4a) may have played an important role in proving the author's newly revised principle of generating electrical energy.

[Response from Comment 2]

The work function of the electrode is one of the factors that affect the generation of electrical energy, but in the system presented in this study, I think that the activity of the electrode may have played a more important role.

[Response from Comment 3]

Please include this data into the supporting information.

[Response from Comment 4]

The new pH test conducted by the author was a faithful answer to this comment. The modified working principle (Figure 1b) is more accurate than the previous one.

[Response from Comment 5]

I agree with the point mentioned that it is different from the existing HEEG devices through the fact that there is no change in electrical energy in the air flow condition. In addition, I believe that the principle of electrical energy generation by the redox reaction that the author claims would have become more solid.

Overall, the author faithfully responded to the reviewer's comments. Considering that a solid

mechanism was designed from simulations and new experiments performed through the revision process, I believe that the current status of this manuscript is suitable for publication in the journal of "Nature Communications".

Reviewer #3 (Remarks to the Author):

The authors have revised the manuscript, and all concerns have been solved and well-explained. I am satisfied with the response and would suggest acceptance of this paper.

Point-to-point response to the Reviewers' comments on the manuscript

**“Sustainable power generation for at least one month
from ambient humidity using unique nanofluidic diode”**

We are with great pleasure to hear this news and we have carefully revised the corresponding part of the manuscript and supporting information point-to-point according to the reviewers' comments. At the same time, we sincerely thank the reviewers for their comments and suggestions on our research work of this paper.

Reviewer #1

[General comment]

- *Debye screening effect*

The author proved that the surface properties change depending on the presence or absence of moisture from topography, but it is still unknown whether the Debye screening effect actually worked. However, since it is common to introduce the concept of Debye screening length in the research dealing with pore size, the author's approach can be considered a good attempt. Please include this part in the supporting information.

- *Simulation of the movement or distribution of ions within CNTs and AAOs*

The author verified the distribution and movement of ions through the simulation. It is a good attempt to understand and explain the mechanism of HEEG power generation and to clarify the behavior of ions in the HEEG system.

- *Conversion relationship between the ionic and electron currents*

The reversible redox reaction described by the author seems to have broadened the scope of the experiment as it is a reaction in an active metal, and I think this content is very important in this study. Additional evidence (Supplementary Fig. 9 and Supplementary Fig. 10) that the redox reaction of electrode materials contributes to electrical output is reliable data that can explain the mechanism by understanding the correlation between ion current and electron current.

- *Discussing the experimental phenomenon of ion rectification*

In the “Analysis of mechanism and influence factors” part, the author matched the electrical characteristics with forward/reverse bias applied to the device with the simulation data, resulting in a reliable discussion and valid data on the ion rectification phenomenon, which is the core of this study. Through this, the content of the manuscript has become more robust.

[Response] Thanks to the reviewer's comments and suggestions. We have added a piece of text: “Although the experimental results have proved that liquid water does

exist on the surface of the device morphology in a humid environment, it is still unknown whether the Debye screening effect plays a role in power generation. However, since it is common to introduce the concept of Debye screening length in the research dealing with pore size, we still use this concept of Debye screening effect in the manuscript.” below the supplementary figure 1e and 1f.

[Comment 1] *It is impressive that the author provided a clear reason for using the In-Ga electrode. I believe that the output performance measured by different metal materials for the bottom electrode (Figure 4a) may have played an important role in proving the author's newly revised principle of generating electrical energy.*

[Response] We thank reviewer's comments and helpful suggestions.

[Comment 2] *The work function of the electrode is one of the factors that affect the generation of electrical energy, but in the system presented in this study, I think that the activity of the electrode may have played a more important role.*

[Response] Thanks to the reviewer's comments. We agree with the view that "electrode activity has more effects on electrical energy generation". Indeed, as shown in the supplementary tablet 4, metal electrodes with different activities have different standard electrode potentials, and the experimental results in Figure 4a also prove that metals with greater activity have better electrical output performance.

[Comment 3] *Please include this data into the supporting information.*

[Response] Thanks to the reviewer's reminding and suggestion. We have put the corresponding experimental results in the Supplementary Figure 19 of the supplementary information.

[Comment 4] *The new pH test conducted by the author was a faithful answer to this comment. The modified working principle (Figure 1b) is more accurate than the previous one.*

[Response]: Thanks for your comments and suggestions.

[Comment 5] *I agree with the point mentioned that it is different from the existing HEEG devices through the fact that there is no change in electrical energy in the air flow condition. In addition, I believe that the principle of electrical energy generation by the redox reaction that the author claims would have become more solid.*

[Response]: Thanks for your comments and suggestions.

Overall, the author faithfully responded to the reviewer's comments. Considering that a solid mechanism was designed from simulations and new experiments performed through the revision process, I believe that the current status of this manuscript is suitable for publication in the journal of "Nature Communications".

[Response]: Thank you for the comments, we are very glad to hear this news.

Reviewer #3

[General comment] *The authors have revised the manuscript, and all concerns have been solved and well-explained. I am satisfied with the response and would suggest acceptance of this paper.*

[Response] We are glad that the reviewer has recognized our revision work, and thank the reviewer for your comments and suggestions on our research work of this paper.